# Less Greedy Equivalence Search

**Adiba Ejaz**      **Elias Bareinboim**
Causal Artificial Intelligence Lab
Columbia University
{adiba.ejaz, eb}@cs.columbia.edu

## Abstract

Greedy Equivalence Search (GES) is a classic score-based algorithm for causal discovery from observational data. In the sample limit, it recovers the Markov equivalence class of graphs that describe the data. Still, it faces two challenges in practice: computational cost and finite-sample accuracy. In this paper, we develop Less Greedy Equivalence Search (LGES), a variant of GES that retains its theoretical guarantees while partially addressing these limitations. LGES modifies the greedy step; rather than always applying the highest-scoring insertion, it avoids edge insertions between variables for which the score implies some conditional independence. This more targeted search yields up to a 10-fold speed-up and a substantial reduction in structural error relative to GES. Moreover, LGES can guide the search using prior knowledge, and can correct this knowledge when contradicted by data. Finally, LGES can use interventional data to refine the learned observational equivalence class. We prove that LGES recovers the true equivalence class in the sample limit, even with misspecified knowledge. Experiments demonstrate that LGES outperforms GES and other baselines in speed, accuracy, and robustness to misspecified knowledge. Our code is available at https://github.com/CausalAILab/lges.

## 1 Introduction

Causal discovery, the task of learning causal structure from data, is a core problem in the field of causality [54]. The causal structure may be an end in itself to the scientist, or a prerequisite for downstream tasks such as inference, decision-making, and generalization [2, 42]. Causal discovery algorithms have been used in a range of disciplines that span biology, medicine, climate science, and neuroscience, among others [19, 43, 47, 48].

A hallmark algorithm in the field is Greedy Equivalence Search (GES) [9, 35], which takes as input observational data and finds a *Markov equivalence class* (MEC) of causal graphs that describe the data. In general, the true graph is not uniquely identifiable from observational data, and the MEC is the most informative structure that can be learned. Under standard assumptions in causal discovery, GES is guaranteed to recover the true MEC in the sample limit. In contrast, many causal discovery algorithms—including prominent examples such as max-min hill-climbing [56] and NoTears [61]—lack such large-sample guarantees. Many variants of GES have been developed, including faster, parallelized implementations [43], restricted search over bounded in-degree graphs [10], and Greedy Interventional Equivalence Search (GIES) [24]. The last of these can exploit interventional data, though is not asymptotically correct [58].

Despite its attractive features, the GES family faces challenges shared across most causal discovery algorithms. For instance, the problem of causal discovery is NP-hard [11], and GES commonly struggles to scale in high-dimensional settings. Moreover, in finite-sample regimes, GES often fails to recover the true MEC. In other words, applying GES in practice is challenging due to both

39th Conference on Neural Information Processing Systems (NeurIPS 2025).

computational complexity (scaling) and sample complexity (accuracy) issues. We refer readers to [56] for an extensive empirical study of GES performance.

At a high level, GES searches over the space of MECs by inserting and deleting edges to maximize a score reflecting data fit. At each state, it evaluates a set of neighbors—possibly exponentially many—and moves to the *highest-scoring* neighbor that scores more than the current MEC. It continues the search greedily until no higher-scoring neighbors are found. In the sample limit, this strategy is guaranteed to find the global optimum of the score: the true MEC.

In this paper, we question the basic assumption of the GES family that the greedy choice of the highest-scoring neighbor is the best one. We first introduce *Generalized GES* (Alg. 3), which allows moving to any score-increasing neighbor of the current MEC, not necessarily the highest-scoring one. This relaxed strategy still finds the global optimum of the score in the sample limit (Thm. 1). More importantly, it opens the door to more strategic neighbor selection.

While it may seem that choosing the highest-scoring neighbor would yield the best performance in practice, surprisingly, we show that this is not the case; a careful and less greedy choice improves both accuracy and runtime. Specializing GGES, we develop the algorithm *Less Greedy Equivalence Search* (LGES) (Alg. 1), advancing on GES and its relatives in the following ways:

1. **Faster, more accurate structure learning.** In Sec. 3.2, we introduce two novel operator selection strategies, CONSERVATIVEINSERT and SAFEINSERT, which LGES exploits for neighbor selection. Empirically, these procedures yield up to a 10-fold reduction in runtime and 2-fold reduction in structural error relative to GES (Experiment 5.1) and other baselines. LGES with SAFEINSERT asymptotically recovers the true MEC (Prop. 2, Cor. 1).

2. **Robustness to misspecified prior knowledge.** In Sec. 3.3, we propose a method whereby LGES can guide neighbor selection based on prior knowledge (Alg. 6) while remaining asymptotically correct even if the knowledge is misspecified. Accurate knowledge improves LGES' accuracy and runtime; inaccurate knowledge harms LGES significantly less than it does GES initialized with the knowledge (Experiment 5.2).

3. **Refining the learned MEC with interventional data.** In Sec. 4, we develop $\mathcal{I}$-ORIENT (Alg. 2, Thm. 2), a score-based procedure that LGES (or any structure learning algorithm) can use to refine an observational MEC with interventional data. To our knowledge, this is the first asymptotically correct score-based procedure for learning from interventional data that can scale to graphs with more than a hundred nodes. LGES with $\mathcal{I}$-ORIENT is 10x faster than GIES [24] while maintaining competitive accuracy (Experiment 5.3).

Proofs for all results are provided in Appendix C. Additional experiments with synthetic data and real-world protein signalling data [48] are provided in Appendix D.

## 2  Background

**Notation.** Capital letters denote variables ($V$), small letters denote their values ($v$), and bold letters denote sets of variables ($\mathbf{V}$) and their values ($\mathbf{v}$). $P(\mathbf{v})$ denotes a probability distribution over a set of variables $\mathbf{V}$. For disjoint sets of variables $\mathbf{X}, \mathbf{Y}, \mathbf{Z}$, $\mathbf{X} \perp\!\!\!\perp \mathbf{Y} \mid \mathbf{Z}$ denotes that $\mathbf{X}$ and $\mathbf{Y}$ are conditionally independent given $\mathbf{Z}$ and $\mathbf{X} \perp_d \mathbf{Y} \mid \mathbf{Z}$ denotes that $\mathbf{X}$ and $\mathbf{Y}$ are *d-separated* given $\mathbf{Z}$ in the graph in context.

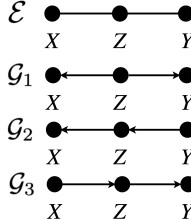

**Causal graphs [3, 42].** A causal graph over variables $\mathbf{V}$ is a directed acyclic graph (DAG) with an edge $X \to Y$ denoting that $X$ is a possible cause of $Y$. The parents of a variable $X$ in a graph $\mathcal{G}$, denoted $\mathbf{Pa}_X^{\mathcal{G}}$, are those variables with a directed edge into $X$. The non-descendants of a variable $X$ in a graph $\mathcal{G}$, denoted $\mathbf{Nd}_X^{\mathcal{G}}$, are those variables to which there is no directed path from $X$ (excluding $X$ itself). The superscript will be omitted when clear from context. A given distribution $P(\mathbf{v})$ is said to be *Markov* with respect to a DAG $\mathcal{G}$ if for all disjoint sets $\mathbf{X}, \mathbf{Y}, \mathbf{Z} \subseteq \mathbf{V}$, if $\mathbf{X} \perp_d \mathbf{Y} \mid \mathbf{Z}$ in $\mathcal{G}$, then $\mathbf{X} \perp\!\!\!\perp \mathbf{Y} \mid \mathbf{Z}$ in $P(\mathbf{v})$. If the converse is also true, $P(\mathbf{v})$ is said to be *faithful* to $\mathcal{G}$. In this work, like GES [9], we assume

Figure 1: A CPDAG $\mathcal{E}$ and the three DAGs in the MEC it represents, encoding $X \perp_d Y \mid Z$.

that the system of interest is *Markovian*, i.e. it contains no unobserved confounders, and that there exists a DAG $\mathcal{G}$ with respect to which the given $P(\mathbf{v})$ is both Markov and faithful.

**Markov equivalence classes [41, 54].**  Two causal DAGs $\mathcal{G}, \mathcal{H}$ are said to be Markov equivalent if they encode exactly the same $d$-separations. The Markov equivalence class (MEC) of a DAG is the set of all graphs that are Markov equivalent to it. A given $P(\mathbf{v})$ may be Markov and faithful to more than one DAG. Hence, the target of causal discovery from observational data is the MEC of DAGs with respect to which $P(\mathbf{v})$ is Markov and faithful. An MEC $\mathcal{M}$ is represented by a unique completed partially directed graph (CPDAG). A CPDAG $\mathcal{E}$ for $\mathcal{M}$ has an undirected edge $X - Y$ if $\mathcal{M}$ contains two DAGs $\mathcal{G}_1, \mathcal{G}_2$ with $X \rightarrow Y$ in $\mathcal{G}_1$ and $Y \rightarrow X$ in $\mathcal{G}_2$. $\mathcal{E}$ has a directed edge $X \rightarrow Y$ if $X \rightarrow Y$ is in every DAG in $\mathcal{M}$. We frequently refer to an MEC by its representative CPDAG. The adjacencies (neighbours) of a variable $X$ in a CPDAG $\mathcal{E}$, denoted $\mathbf{Adj_X^{\mathcal{G}}}$ ($\mathbf{Ne_X^{\mathcal{G}}}$), comprise those variables connected by any edge (an undirected edge) to $X$.

**Greedy Equivalence Search [9, 35].**  Greedy Equivalence Search (GES) is a score-based algorithm for learning MECs from observational data. It searches for the true MEC by maximizing a scoring criterion given $m$ samples of data $\mathbf{D} \sim P(\mathbf{v})$. For example, a popular choice of scoring criterion is the *Bayesian information criterion* (BIC) [51].

GES assumes that the given scoring criterion is decomposable, consistent, and score-equivalent, so that the score of an MEC is the score of any DAG in that MEC (Defs. A.3, A.4, A.5). BIC satisfies each of these conditions for distributions that are Markov and faithful to some DAG and are curved exponential families, for e.g., linear-Gaussian or multinomial models [9, 21, 23]. Moreover, decomposability and consistency imply local consistency ([9, Lemma 7]), the key property needed for the correctness of GES.

**Definition 1** (Locally consistent scoring criterion [9, Def. 6])**.**  Let $\mathbf{D}$ be a dataset consisting of i.i.d. samples from some distribution $P(\mathbf{v})$. Let $\mathcal{G}$ be any DAG, and let $\mathcal{G}'$ be the DAG that results from adding the edge $X \rightarrow Y$ to $\mathcal{G}$. A scoring criterion $S$ is said to be *locally consistent* if, as the number of samples goes to infinity, the following two properties hold:

1. If $X \not\perp\!\!\!\perp Y \mid \mathbf{Pa}_Y^{\mathcal{G}}$ in $P(\mathbf{v})$ then $S(\mathcal{G}, \mathbf{D}) < S(\mathcal{G}', \mathbf{D})$.

2. If $X \perp\!\!\!\perp Y \mid \mathbf{Pa}_Y^{\mathcal{G}}$ in $P(\mathbf{v})$ then $S(\mathcal{G}, \mathbf{D}) > S(\mathcal{G}', \mathbf{D})$.

**Example 1.**  Consider a distribution $P(\mathbf{v})$ whose true MEC is $\mathcal{E}$ and true DAG is $\mathcal{G}_2 \in \mathcal{E}$ as in Fig. 1. Consider $\mathcal{G}_1 \in \mathcal{E}$ (Fig. 1), and let $\mathcal{G}_1^+ = \mathcal{G}_1 \cup \{X \rightarrow Y\}$ and $\mathcal{G}_1^- = \mathcal{G}_1 \setminus \{Z \rightarrow Y\}$. Since $Y \perp\!\!\!\perp X \mid \mathbf{Pa}_Y^{\mathcal{G}_1}$ in $P(\mathbf{v})$, where $\mathbf{Pa}_Y^{\mathcal{G}_1} = \{Z\}$, $\mathcal{G}_1^+$ has a lower score than $\mathcal{G}_1$. Since $Y \not\perp\!\!\!\perp Z \mid \mathbf{Pa}_Y^{\mathcal{G}_1^-}$ in $P(\mathbf{v})$, where $\mathbf{Pa}_Y^{\mathcal{G}_1^-} = \emptyset$, $\mathcal{G}_1^-$ has a lower score than $\mathcal{G}_1$. $\qquad\square$

Given a scoring criterion satisfying the above conditions and data $\mathbf{D} \sim P(\mathbf{v})$ where $P(\mathbf{v})$ is Markov and faithful to some DAG, GES recovers the true MEC in the sample limit [9, Lemma 10]. The PC algorithm [54], a constraint-based method, has similar asymptotic correctness guarantees, but uses conditional independence (CI) tests instead of a score. PC starts with a fully connected graph and removes edges using CI tests. In contrast, GES starts with a fully disconnected graph and proceeds in two phases. In the forward phase, at each state, GES finds the highest-scoring INSERT operator that results in a score increase, applies it, and repeats until no score-increasing INSERT operator exists. At this point, it has found an MEC $\mathcal{E}$ with respect to which $P(\mathbf{v})$ is Markov.

**Definition 2** (INSERT operator, [9, Def. 12])**.**  Given a CPDAG $\mathcal{E}$, non-adjacent nodes $X, Y$ in $\mathcal{E}$, and some $\mathbf{T} \subseteq \mathbf{Ne}_Y^{\mathcal{E}} \setminus \mathbf{Adj}_X^{\mathcal{E}}$, the INSERT$(X, Y, \mathbf{T})$ operator modifies $\mathcal{E}$ by inserting the edge $X \rightarrow Y$ and directing the previously undirected edges $T - Y$ for $T \in \mathbf{T}$ as $T \rightarrow Y$.

Intuitively, an INSERT$(X, Y, \mathbf{T})$ operator applied to an MEC $\mathcal{E}$ corresponds to choosing a DAG $\mathcal{G} \in \mathcal{E}$ (depending on $X, Y$, and $\mathbf{T}$), adding the edge $X \rightarrow Y$ to $\mathcal{G}$, and computing the MEC of the resulting DAG. Though $P(\mathbf{v})$ is Markov with respect to the MEC $\mathcal{E}$ found in the forward phase, $P(\mathbf{v})$ may not be faithful to $\mathcal{E}$. In the backward phase, GES starts the search with $\mathcal{E}$, finds the highest-scoring DELETE$(X, Y, \mathbf{H})$ operator (Def. A.2) that results in a score increase, applies it, and repeats until no score-increasing DELETE operator exists. At this point, it has found an MEC with respect to which $P(\mathbf{v})$ is both Markov and faithful. An optional turning phase is known to improve performance in practice, but is redundant for asymptotic correctness [24].

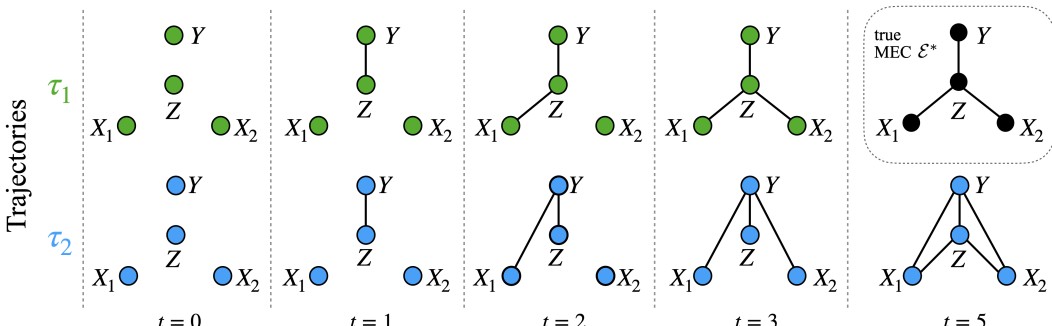

Figure 2: Possible trajectories, $\tau_1$ and $\tau_2$, that GES may take in the forward phase to obtain an MEC with respect to which a given distribution $P(\mathbf{v})$ is Markov. The true MEC is $\mathcal{E}^*$ (top right). In each trajectory, $\mathcal{E}^{(t+1)}$ results from applying some INSERT operator to $\mathcal{E}^{(t)}$.

# 3 Less Greedy Equivalence Search

## 3.1 Generalizing GES

To lay the groundwork for our search strategy, we first introduce Generalized GES (GGES) (Alg. 3), which generalizes GES in three ways. Firstly, GGES allows the search to be initialized from an arbitrary MEC $\mathcal{E}_0$, rather than the empty graph. Secondly, GGES is not restricted to the forward-backward-turning phase structure of GES; it can accommodate any order of operators (e.g., deletions before insertions [36]) as specified by the user. Finally, GGES allows the application of *any* valid score-increasing operator, instead of just the highest-scoring one.

At each state $\mathcal{E}$, GGES calls abstract subroutine GETOPERATOR, which either returns a valid score-increasing operator of the type specified (insertion, deletion, reversal, or either), if one exists, or indicates that there is no such operator. This proceeds until no score-improving operators are found.

**Theorem 1** (Correctness of GGES). *Let $\mathcal{E}$ denote the Markov equivalence class that results from GGES (Alg. 3) initialised from an arbitrary MEC $\mathcal{E}_0$ and let $P(\mathbf{v})$ denote the distribution from which the data $\mathbf{D}$ was generated. Then, as the number of samples goes to infinity, $\mathcal{E}$ is the Markov equivalence class underlying $P(\mathbf{v})$.*

In the next section, we illustrate how GETOPERATOR can be implemented in a way that yields significant improvements in accuracy and runtime relative to GES.

## 3.2 An improved insertion strategy

In practice, the output of GES is known to include adjacencies between many variables that are non-adjacent in the true MEC [36, 56]. Since these adjacencies are introduced by INSERT operators, this motivates a more careful choice of which INSERT operator to apply. Our approach is grounded in the following observation.

**Proposition 1.** Let $\mathcal{E}$ denote an arbitrary CPDAG and let $P(\mathbf{v})$ denote the distribution from which the data $\mathbf{D}$ was generated. Assume, as the number of samples goes to infinity, that there exists a valid score-decreasing INSERT$(X, Y, \mathbf{T})$ operator for $\mathcal{E}$. Then, there exists a DAG $\mathcal{G} \in \mathcal{E}$ such that (1) $Y \perp_d X \mid \mathbf{Pa}_Y^{\mathcal{G}}$ and (2) $Y \perp\!\!\!\perp X \mid \mathbf{Pa}_Y^{\mathcal{G}}$ in $P(\mathbf{v})$.

Then, for variables $X$ and $Y$, even a single score-decreasing INSERT$(X, Y, \mathbf{T})$ implies that $X$ and $Y$ are non-adjacent in the true MEC. However, this does not imply that all INSERT$(X, Y, *)$ are also score-decreasing. GES may thus apply a different INSERT$(X, Y, \mathbf{T}')$, introducing an adjacency not present in the true MEC. The following example shows how such choices can lead GES to MECs that contain many excess adjacencies.

**Example 2.** Consider a distribution $P(\mathbf{v})$ over $\mathbf{V} = \{X_1, X_2, Y, Z\}$ whose true MEC is given by $\mathcal{E}^*$ in Fig. 2 (top right). GES starts with the empty graph and successively applies the highest-scoring INSERT operator that it finds. Trajectories $\tau_1$ and $\tau_2$ agree until time $t = 1$. Let $\mathcal{E}^{(1)}$ denote the CPDAG common to $\tau_1$ and $\tau_2$ at $t = 1$. At $t = 1$, GES has many INSERT operators it could apply to

$\mathcal{E}^{(1)}$. Recall that each $\text{INSERT}(\alpha, \beta, \mathbf{T})$ applied to $\mathcal{E}^{(1)}$ corresponds to choosing some DAG $\mathcal{G}$ from $\mathcal{E}^{(1)}$ and adding $\alpha \to \beta$ to it. The DAG $\mathcal{G}$ is chosen such that for edges $\gamma - \beta$ in $\mathcal{E}^{(1)}$ where $\alpha$ and $\gamma$ are non-adjacent, $\mathcal{G}$ contains $\gamma \to \beta$ if $\gamma \in \mathbf{T}$ and $\beta \to \gamma$ otherwise.

1. $\alpha = X_1, \beta = Z, \mathbf{T} = \emptyset$. This corresponds to choosing $\mathcal{G}_1 \in \mathcal{E}^{(1)}$ (which already has $Z \to Y$) and adding $X_1 \to Z$ to it (Fig. 3, left). Since $Z \not\perp\!\!\!\perp X_1 \mid \mathbf{Pa}_Z^{\mathcal{G}_1}$, this edge addition increases the score of $\mathcal{G}_1$ (by local consistency, Def. 1) and hence of $\mathcal{E}^{(1)}$. This operator is chosen in trajectory $\tau_1$.

2. $\alpha = X_1, \beta = Y, \mathbf{T} = \{Z\}$. This corresponds to choosing $\mathcal{G}_1 \in \mathcal{E}^{(1)}$ (which already has $Z \to Y$) and adding $X_1 \to Y$ to it (Fig. 3, middle). Since $Y \perp\!\!\!\perp X_1 \mid \mathbf{Pa}_Y^{\mathcal{G}_1}$, this edge addition decreases the score of $\mathcal{G}_1$ and hence of $\mathcal{E}^{(1)}$. This operator is never chosen.

3. $\alpha = X_1, \beta = Y, \mathbf{T} = \emptyset$. This corresponds to choosing $\mathcal{G}_2 \in \mathcal{E}^{(1)}$ (which already has $Y \to Z$) and adding $X_1 \to Y$ to it (Fig. 3, right). Since $Y \not\perp\!\!\!\perp X_1 \mid \mathbf{Pa}_Y^{\mathcal{G}_2}$, this edge addition increases the score of $\mathcal{G}_1$ and hence of $\mathcal{E}^{(1)}$. This operator is chosen in trajectory $\tau_2$.

In the sample limit, it is unknown whether $\mathcal{G}_A$ or $\mathcal{G}_C$ would score higher. For an extended discussion, see Ex. B.1. [1]  □

The above example shows that GES may insert an edge between non-adjacent variables even in simple settings with a small number of variables. Such choices accumulate in higher-dimensional settings. This motivates avoiding edge insertions for variable pairs $(X, Y)$ for which a score-decreasing $\text{INSERT}$ is observed. We hypothesize this has two benefits: (1) *accuracy*: it avoids inserting excess adjacencies that the backward

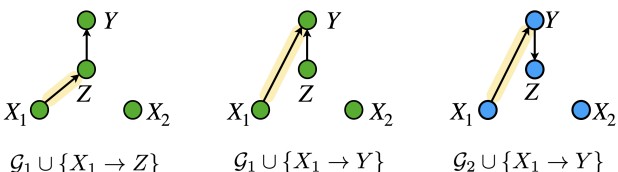

$\mathcal{G}_1 \cup \{X_1 \to Z\}$     $\mathcal{G}_1 \cup \{X_1 \to Y\}$     $\mathcal{G}_2 \cup \{X_1 \to Y\}$

Figure 3: Illustration of some $\text{INSERT}$ operators that may be applied to the MEC $\mathcal{E}^{(1)}$ at $t = 1$ in Fig. 2. These operators correspond to various edge additions to the DAGs $\mathcal{G}_1, \mathcal{G}_2 \in \mathcal{E}^{(1)}$, where $\mathcal{G}_1$ orients $Z - Y$ as $Z \to Y$ and $\mathcal{G}_2$ orients $Z - Y$ as $Y \to Z$.

phase may fail to remove, and (2): *efficiency*: it stops the enumeration of $(X, Y)$ insertions when a lower-scoring one is found; moreover, reducing excess adjacencies reduces the number of operators that need to be evaluated in subsequent states.

We now formalize two strategies for avoiding such insertions.

**Strategy 1** (CONSERVATIVEINSERT). At a given state with CPDAG $\mathcal{E}$, for each non-adjacent pair $(X, Y)$, iterate over valid $\text{INSERT}(X, Y, \mathbf{T})$. If any score-decreasing $\mathbf{T}$ is found, stop, discard all $\text{INSERT}(X, Y, *)$ operators and continue to the next pair. Among all retained candidates, select the highest-scoring operator that results in a score increase, if any.

CONSERVATIVEINSERT avoids inserting edges between any variables $(X, Y)$ for which some conditional independence has been found, as evidenced by a score-decreasing $\text{INSERT}$ (Prop. 1). While intuitive, it is unknown if this strategy is guaranteed to find a score-increasing $\text{INSERT}$ whenever one exists. To elaborate, the soundness of CONSERVATIVEINSERT rests on the following premise: if $P(\mathbf{v})$ is not Markov with respect to an MEC $\mathcal{E}$, then there must exist variables $X, Y$ such that for every $\mathcal{G} \in \mathcal{E}$, $X \not\perp\!\!\!\perp Y \mid \mathbf{Pa}_Y^{\mathcal{G}}$. Importantly, the choice of $X, Y$ can not depend on $\mathcal{G}$. This is the main challenge in proving the soundness of CONSERVATIVEINSERT. Still, we provide partial guarantees in Prop. C.1, C.2, but leave the soundness of CONSERVATIVEINSERT open.

Furthermore, we introduce SAFEINSERT, a relaxation of CONSERVATIVEINSERT that is guaranteed to find a score-increasing $\text{INSERT}$ when one exists. The soundness of SAFEINSERT only requires that

---

[1] We can also ask, which of these $\text{INSERT}$ operators scores the highest in practice? We generated 100 linear-Gaussian datasets of 100 samples each according to a fixed true DAG in $\mathcal{E}^*$, following the set-up in Sec. 5.1. Then, we computed the scores of $\mathcal{G}_A : \mathcal{G}_1 \cup \{X_1 \to Z\}$, $\mathcal{G}_B : \mathcal{G}_1 \cup \{X_1 \to Y\}$, and $\mathcal{G}_C : \mathcal{G}_2 \cup \{X_1 \to Y\}$ on each dataset. From the fact that $\mathcal{G}_A$ is closer to the true MEC than $\mathcal{G}_C$, it may seem that $\mathcal{G}_A$ would always score higher. However, $\mathcal{G}_A$ was the highest-scoring DAG 96% of the time, and $\mathcal{G}_C$ 4% of the time. As expected, $\mathcal{G}_B$ is never the highest-scoring DAG.

---

**Algorithm 1:** Less Greedy Equivalence Search (LGES)

---

**Input:** Data $\mathbf{D} \sim \mathbf{P}(\mathbf{v})$, scoring criterion $S$, prior assumptions $\mathbf{S} = \langle \mathbf{R}, \mathbf{F} \rangle$, initial MEC $\mathcal{E}_0$,
      insertion strategy $GetInsert$ in {GETSAFEINSERT, GETCONSERVATIVEINSERT}

**Output:** MEC $\mathcal{E}$ of $\mathbf{P}(\mathbf{v})$

1   $\mathcal{E} \leftarrow \mathcal{E}_0$ ;                    `// allows initialisation if preferred by user`

2   **repeat**

3     **repeat**

4       $\mathcal{E} \leftarrow \mathcal{E}+$ the highest-scoring DELETE$(X, Y, \mathbf{T})$

5     **until** *no score-increasing deletions exist*;

6     **repeat**

7       $\mathcal{E} \leftarrow \mathcal{E}+$ the highest-scoring TURN$(X, Y, \mathbf{T})$

8     **until** *no score-increasing reversals exist*;

9     $\mathcal{G} \leftarrow$ some DAG in $\mathcal{E}$;

10    $priorityList \leftarrow$ GETPRIORITYINSERTS$(\mathcal{E}, \mathcal{G}, \mathbf{S})$;

11    **foreach** *candidates in priorityList* **do**

12      $(X_{max}, Y_{max}, \mathbf{T}_{max}) \leftarrow GetInsert(\mathcal{E}, \mathcal{G}, \mathbf{D}, candidates, S)$;

13      **if** $(X_{max}, Y_{max}, \mathbf{T}_{max})$ *is found* **then**

14        $\mathcal{E} \leftarrow \mathcal{E} +$ INSERT$(X_{max}, Y_{max}, \mathbf{T}_{max})$;

15        break ;              `// no need to check lower priority`

16   **until** *no score-increasing operators exist*;

17   **return** $\mathcal{E}$

---

if $P(\mathbf{v})$ is not Markov with respect to ,$\mathcal{E}$, then for every $\mathcal{G} \in \mathcal{E}$, there must exist variables $X, Y$ such that $X \not\perp\!\!\!\perp Y \mid \mathbf{Pa}_Y^{\mathcal{G}}$. This is unlike CONSERVATIVEINSERT, where $X, Y$ can not depend on $\mathcal{G}$.

**Strategy 2** (SAFEINSERT). At a given state with CPDAG $\mathcal{E}$, pick an arbitrary DAG $\mathcal{G} \in \mathcal{E}$. For each non-adjacent pair $(X, Y)$ in $\mathcal{G}$, check if $\mathcal{G}$ has a higher score than $\mathcal{G} \cup \{X \rightarrow Y\}$. If so, discard all INSERT$(X, Y, *)$ operators and continue to the next pair. Among all retained candidates, select the highest-scoring operator that results in a score increase, if any.

**Proposition 2** (Correctness of SAFEINSERT). *Let $\mathcal{E}$ denote a Markov equivalence class and let $P(\mathbf{v})$ denote the distribution from which the data $\mathbf{D}$ was generated. Then, as the number of samples goes to infinity, SAFEINSERT returns a valid score-increasing INSERT operator if and only if one exists.*

**Example 3.** (Ex. 2 continued). Let $\mathcal{E}^{(1)}, \mathcal{G}_1$, and $\mathcal{G}_2$ be as in Ex. 2. Assume GES is at $\mathcal{E}^{(1)}$ and SAFEINSERT picks the DAG $\mathcal{G}_1 \in \mathcal{E}$. Then, $\mathcal{G}_1 \cup \{X_1 \rightarrow Y\}$ has a lower score than $\mathcal{G}_1$ since $X_1 \perp\!\!\!\perp Y \mid \mathbf{Pa}_Y^{\mathcal{G}_1}$ in $P(\mathbf{v})$, where $\mathbf{Pa}_Y^{\mathcal{G}_1} = \{Z\}$. SAFEINSERT thus does not consider any INSERT$(X_1, Y, *)$ operators. In contrast, assume SAFEINSERT picks the DAG $\mathcal{G}_2 \in \mathcal{E}$. Then, $\mathcal{G}_2 \cup \{X_1 \rightarrow Y\}$ has a higher score than $\mathcal{G}_2$, and SAFEINSERT may still consider INSERT$(X_1, Y, *)$ operators. However, CONSERVATIVEINSERT will not consider any INSERT$(X_1, Y, *)$ operators, since INSERT$(X_1, Y, \{Z\})$, corresponding to $\mathcal{G}_1 \cup \{X_1 \rightarrow Y\}$, results in a lower score than $\mathcal{E}^{(1)}$.     □

Pseudocode for the above insertion strategies are in Algs. 4 and 5. Later, in Sec. 5.1, we compare these strategies, and show how both achieve substantial gains in accuracy and runtime over GES.

### 3.3   Learning with prior knowledge

In this section, we present another modification of the forward phase of GES: prioritizing edge insertions based on an expert's prior causal knowledge. The insight that underpins GGES—that we can apply *any* score-increasing insertion—suggests a new way to incorporate such assumptions while still correcting them if contradicted by the data. We assume that we are given a possibly misspecified causal model as a set of required and forbidden edges $\mathbf{S} = \langle \mathbf{R}, \mathbf{F} \rangle$ that may be directed or undirected.

**Initialization from prior assumptions.** A natural strategy, which we refer to as GES-INIT, initializes the search to an MEC consistent with the assumptions, for e.g., by including all edges in $\mathbf{R}$, and then proceeds greedily as in standard GES.[2] This approach, an instantiation of GGES, is sound in the large-sample limit (as a corollary of Thm. 1), even when the assumptions are misspecified.

---

[2]This was empirically evaluated in [14]. However, its correctness was not considered.

However, given finite samples, such initialisation may harm both accuracy and runtime. If the expert suggests adjacencies that don't exist in the true MEC, GES-INIT includes them by default in the initialisation and may fail to remove them later. Moreover, such initialisation precludes the use of insertion strategies from Sec. 3.2 that would avoid introducing such excess adjacencies.

**Guided search from prior assumptions.** We instead propose a strategy that uses prior assumptions to *prioritize* operators, and not to initialize the search. Specifically, for each non-adjacent pair $(X, Y)$, we rank it into one of four categories based on the constraint set $\mathbf{S} = \langle \mathbf{R}, \mathbf{F} \rangle$ using the procedure GETPRIORITYINSERTS (Alg. 6). Insertions for higher-priority adjacencies are considered first, but only applied if they increase the score. For example, if SAFEINSERT finds no score-increasing insertions for the current MEC, then the remaining expert-provided edges in $\mathbf{R}$ (if any) are not consistent with the data, and will not be inserted. In contrast, GES-INIT inserts all edges by default.

Next, in Sec. 3.4, we incorporate this prioritization scheme into a novel algorithm, combining it with the search strategy of Sec. 3.2 to enable a less greedy search. In Sec. 5.2, we empirically demonstrate the benefit of this prioritization-based strategy.

### 3.4 The Less Greedy Equivalence Search algorithm

Finally, we introduce the main result of this work: the algorithm Less Greedy Equivalence Search (LGES, Alg. 1). We present three variants:

1. **LGES-0** (Alg. 8), which modifies the insertion step of GES based on our insights in the previous sections, while using the same search strategy as GES in the deletion step,

2. **LGES** (Alg. 1), which is similar to (1) but additionally incorporates the XGES-0 heuristic of prioritizing insertions before deletions [36], and

3. **LGES+** (Alg. 9), which is similar to (2) but additionally incorporates the XGES heuristic of forcing edge deletions and restarting the search [36].[3]

As a corollary of Thm. 1 and Prop. 2, we can show that each LGES variant with SAFEINSERT recovers the true MEC in the sample limit, even given a misspecified set of prior assumptions.

**Corollary 1** (Correctness of LGES). *Let $\mathcal{E}$ denote the Markov equivalence class that results from LGES (Alg. 1) initialized from an arbitrary MEC $\mathcal{E}_0$ and given prior assumptions $\mathbf{S} = \langle \mathbf{R}, \mathbf{F} \rangle$, and let $P(\mathbf{v})$ denote the distribution from which the data $\mathbf{D}$ was generated. Then, as the number of samples goes to infinity, $\mathcal{E}$ is the Markov equivalence class underlying $P(\mathbf{v})$.*

**Remark 1.** While we only show that LGES with SAFEINSERT is asymptotically correct, LGES can also be run with CONSERVATIVEINSERT. Since we only have partial guarantees on CONSERVATIVEINSERT (Prop. C.1, C.2), it remains open whether this variant of LGES is asymptotically correct.

## 4 Score-based learning from interventional data

While the previous sections address causal discovery from observational data, such data alone leaves many edges unoriented. Interventional data can resolve such ambiguities. In this section, we extend the LGES method with $\mathcal{I}$-ORIENT, a score-based procedure for refining an observational MEC with interventional data. Unlike existing score-based methods, which are asymptotically inconsistent or computationally infeasible even in moderate-dimensional settings [24, 58], our approach scales while preserving soundness.

**Background on interventions.** Following [24], we assume soft unconditional interventions, including hard (do) interventions as a special case. These set the distribution of a variable $X$ to some fixed $P^*(x)$, thereby removing the influence of its parents. Let $\mathcal{I}$ denote a family of interventional targets, i.e., subsets $\mathbf{I} \subseteq \mathbf{V}$, with the empty intervention $\theta \in \mathcal{I}$ producing the observational distribution. We observe data from distributions $(\mathbf{P_I}(\mathbf{v}))_{\mathbf{I} \in \mathcal{I}}$. As in the observational case, we assume there exists a DAG $\mathcal{G}$ such that these distributions are $\mathcal{I}$-Markov (Def. A.7) and faithful to the corresponding intervention graphs $(\mathcal{G}_{\bar{\mathbf{I}}})_{\mathbf{I} \in \mathcal{I}}$, obtained by removing edges into any intervened variable $V \in \mathbf{I}$ [16, 42].

---

[3]Pseudocode and correctness results for LGES-0 and LGES+ can be found in Appendix C.

**Algorithm 2:** $\mathcal{I}$-ORIENT

---

**Input:** Intervention targets $\mathcal{I}$, data $(\mathbf{D_I})_{\mathbf{I} \in \mathcal{I}} \sim (\mathbf{P_I}(\mathbf{v}))_{\mathbf{I} \in \mathcal{I}}$, observational MEC $\mathcal{E}$, scoring
      criterion $S$
**Output:** $\mathcal{I}$-MEC $\mathcal{E}$ of $(\mathbf{P_I}(\mathbf{v}))_{\mathbf{I} \in \mathcal{I}}$

**1 foreach** $X \in \mathcal{E}$ *and* $Y \in ne_X^{\mathcal{E}}$ **do**
**2**     $\Delta S \leftarrow \sum\limits_{\mathbf{I} \in \mathcal{I}, \ X \in \mathbf{I}, Y \notin \mathbf{I}} s_{\mathbf{D_I}}(y, x) - s_{\mathbf{D_I}}(y)$ ;
**3**     **if** $\Delta S > 0$ **then**
**4**        Orient edge $X - Y$ as $X \to Y$ in $\mathcal{E}$ ;
**5**        Apply Meek's rules in $\mathcal{E}$ to propagate orientations [35] ;
**6**     **else if** $\Delta S < 0$ **then**
**7**        Orient edge $X - Y$ as $X \leftarrow Y$ in $\mathcal{E}$ ;
**8**        Apply Meek's rules in $\mathcal{E}$ to propagate orientations [35] ;
**9 return** $\mathcal{E}$

---

Just as observational data identifies an MEC, interventional data identifies an $\mathcal{I}$-MEC, a smaller equivalence class encoding constraints on both observational and interventional data [24, Def. 7].

**Orientation procedure.** To recover the $\mathcal{I}$-MEC, we introduce $\mathcal{I}$-ORIENT (Alg. 2), which orients undirected edges in the observational MEC using scores from interventional data. The underlying idea is simple. Consider some undirected edge $X - Y$ in the observational MEC $\mathcal{E}$, where the ground truth DAG is $\mathcal{G}^*$. Say there exists an intervention $\mathbf{I}$ containing $X$ but not $Y$. When we perform an unconditional intervention on $X$, we erase the causal influence of $\mathbf{Pa}_X^{\mathcal{G}}$ on $X$. So, $Y$ is a parent of $X$ in $\mathcal{G}^*$ if and only if $X$ and $Y$ are marginally independent in $P_{\mathbf{I}}(\mathbf{v})$. How can we check for marginal independence using a scoring criterion? The key is to use local consistency (Def. 1): we compare a graph $\mathcal{G}$ where $Y$ has no parents to a graph $\mathcal{G} \cup \{X \to Y\}$. The former will score higher than the latter if and only if $X \perp\!\!\!\perp Y$ in $P_{\mathbf{I}}(\mathbf{v})$. This is precisely the test in line 2 of $\mathcal{I}$-ORIENT (Alg. 2).

**Theorem 2** (Correctness of $\mathcal{I}$-ORIENT). *Let $\mathcal{E}$ denote the Markov equivalence class that results from $\mathcal{I}$-ORIENT (Alg. 2) given an observational MEC $\mathcal{E}_0$ and interventional targets $\mathcal{I}$, and let $(\mathbf{P_I}(\mathbf{v}))_{\mathbf{I} \in \mathcal{I}}$ denote the family of distributions from which the data $(\mathbf{D_I})_{\mathbf{I} \in \mathcal{I}}$ was generated. Assume that $\mathcal{E}_0$ is the MEC underlying $P_\emptyset(\mathbf{v})$. Then, as the number of samples goes to infinity for each $\mathbf{I} \in \mathcal{I}$, $\mathcal{E}$ is the $\mathcal{I}$-Markov equivalence class underlying $(\mathbf{P_I}(\mathbf{v}))_{\mathbf{I} \in \mathcal{I}}$.*

## 5 Experiments

### 5.1 Learning from observational data

**Synthetic data and baselines.** We draw Erdős–Rényi graphs with $p$ variables and $\{1, 2, 3\} \cdot p$ edges in expectation, denoted ER-$\{1, 2, 3\}$ respectively). We run most experiments for $p$ up to $150$, with additional experiments for $p$ up to $250$ in Sec. D.1.6. For each $p$, we sample 50 graphs and generate linear-Gaussian data for each graph. Following [37], we draw weights from $\mathcal{U}([-2, -0.5] \cup [0.5, 2])$ and noise variances from $\mathcal{U}([0.1, 0.5])$. We obtain samples of size $n \in \{10^3, 10^4\}$ via `sempler` [20]. We evaluate GES, XGES-0 [36], XGES [36], LGES-0, LGES, LGES+, PC [54], and NoTears [61]. [4]

**Results.** LGES (both Safe and Conservative) significantly outperforms XGES and GES in runtime and accuracy. We measure accuracy by Structural Hamming Distance (SHD) [56] between the estimated and true CPDAGs (Fig. 4a), as well as precision, recall, and F1 score (Fig. D.1.2). Furthermore, the less greedy variants of GES, XGES-0, and XGES all improve on their respective original algorithms (Figs. 4b, D.1.1, D.1.3). Of the GES variants, LGES+ (Conservative) is the most accurate, whereas LGES (Conservative) is the fastest. CONSERVATIVEINSERT outperforms SAFEINSERT in accuracy, but both strategies yield similar runtime.

---

[4]All implementations in Python. GES variants are implemented by modifying the code in `https://github.com/juangamella/ges`. We use the PC implementation in `causal-learn` [62] and NoTears implementation in `causal-nex` [5]. Our GES implementations share as much code as possible. We do not use the optimized implementation of the operators proposed in [36] across any of our GES variants, to study the effect of the search strategy independent of implementation.

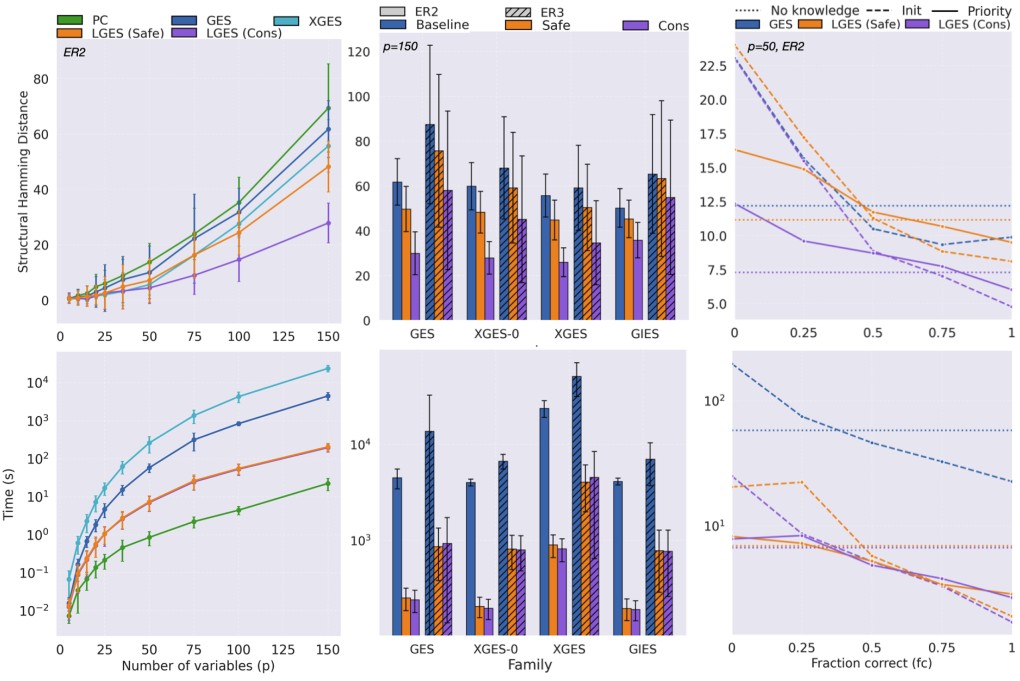

(a) Observational data performance    (b) Impact of less greedy insertion    (c) Impact of knowledge

Figure 4: Performance of algorithms on $50$ simulated datasets from Erdős–Rényi graphs with $p$ variables. **Lower is better** (more accurate / faster) across all plots. **(a)** LGES outperforms baselines in accuracy and runtime on graphs with $2p$ edges in expectation, given $n = 10^4$ observational samples and no prior knowledge. **(b)** Less greedy insertion improves several GES variants on graphs with $p = 150$ variables and $2p$ and $3p$ edges in expectation, given $n = 10^4$ observational samples (and $n = 10^3$ samples per intervention for GIES) and no prior knowledge. LGES-0, LGES, LGES+, and LGIES are the less greedy variants of GES, XGES-0, XGES, and GIES respectively. **(c)** Given prior knowledge in the form of $3m/4$ required edges when the true graph contains $m$ edges, LGES' prioritization strategy is more robust to misspecification in the knowledge than initialization with the same knowledge, given $n = 10^3$ observational samples. See Sec. D for additional results.

To elaborate, LGES (Safe and Conservative) is an order of magnitude faster than GES. LGES (Conservative) is up to 2 times more accurate than GES, for instance, resulting in only $\approx 30$ incorrect edges on average in graphs with $150$ variables and $300$ edges in expectation. These comparisons also hold for LGES-0, the less greedy variant of GES (Figs. 4b, D.1.1, D.1.3). The difference in accuracy is due to excess adjacencies and incorrect orientations; missing adjacencies almost never occur (Fig. D.1.1). PC, though the fastest algorithm, is less accurate than GES for $n = 10^4$, but performs best in the case of many variables and few samples (Fig. D.1.4). NoTears has much worse accuracy than other methods, e.g., average SHD $\approx 125$ on graphs with 100 variables, though its runtime appears to scale better (Fig. D.1.1). See Sec. D.1 for additional results and discussion.

## 5.2 Learning with prior knowledge

**Synthetic data and baselines.** We study how correctness of prior knowledge affects performance when data is limited ($n = 10^3$) on ER2 graphs with 50 variables, with data generated as in Sec. 5.1. For a true DAG $\mathcal{G}$ with $m$ edges, we generate prior assumptions on $m' \in \{m/2,\ 3m/4\}$ required edges as follows. We vary the fraction `fc` of the chosen $m'$ edges that is 'correct', with $c \cdot m'$ edges chosen correctly from those in $\mathcal{G}$ and the remaining chosen incorrectly from those not in $\mathcal{G}$. We compare GES (with and without initialisation) and LGES (without initialisation, with initialisation only, and with priority insertion).

**Results.** Fig. 4c summarizes results for $m' = 3m/4$, with additional results in Sec. D.2. LGES (Conservative) outperforms GES across all levels of prior correctness in terms of time and SHD.

First, we compare initialization with prioritization. When the knowledge is majority incorrect (`fc` $\in \{0.5, 0.25, 0.0\}$), the prioritization strategy significantly outperforms initialization in runtime and accuracy. When the knowledge is mostly accurate (`fc` $\in \{0.75, 1\}$), initialization yields marginally better accuracy than prioritization. Second, we consider when knowledge improves performance. When the knowledge is largely correct (`fc` $\geq 0.75$), it marginally improves the accuracy of LGES with prioritization, and more visibly improves that of LGES with initialisation. However, as long as `fc` $\geq 0.5$, knowledge significantly improves the runtime of LGES with prioritization. Thus, our prioritization strategy (Sec. 3.3) can leverage knowledge while being robust to misspecification.

## 5.3 Learning from interventional data

**Synthetic data and baselines**. We follow a similar set-up as Sec. 5.1 with $10^4$ observational samples and ER-$\{2, 3\}$ graphs. For a graph on $p$ variables, we randomly construct $|\mathcal{I}| = p/10$ interventions and generate $10^3$ samples for each. We compare GIES [24]; LGES run on observational data followed by $\mathcal{I}$-ORIENT; LGIES-0; and LGIES. LGIES-0 and LGIES incorporate less greedy insertion into GIES; the latter also prioritizes deletions before insertions.

**Results.** All less-greedy algorithms are up to $10\times$ faster than GIES (Figs. 4b, 5), with LGIES (Conservative) being the fastest. LGIES (Conservative) is up to $1.5\times$ more accurate than GIES. In general, all less greedy algorithms are more accurate than GIES with the exception of LGES (Safe) + $\mathcal{I}$-ORIENT, which has competitive accuracy with GIES on ER-3 graphs but performs slightly worse on ER-2 graphs. While being the only combination known to be asymptotically correct, LGES (Safe) + $\mathcal{I}$-

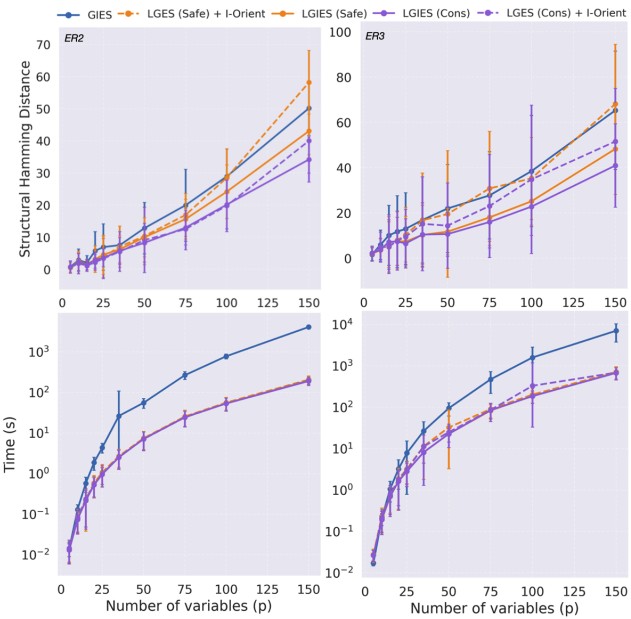

Figure 5: Performance of algorithms on 50 simulated datasets from Erdős–Rényi graphs with $p$ variables and **(left)** $2p$ edges **(right)** $3p$ edges in expectation, given $n = 10^4$ observational samples and $n = 10^3$ samples per intervention (Sec. D.3). **Lower is better** (more accurate / faster) across all plots. LGIES significantly outperforms GIES.

ORIENT is limited by the fact that it uses only observational data to learn the observational MEC, while GIES and LGIES additionally use interventional data to this end.

## 6 Conclusions

In this paper, we introduced a family of less greedy algorithms for score-based causal discovery from observational and interventional data. Our core insight lies in two new operator selection strategies, CONSERVATIVEINSERT and SAFEINSERT, that avoid inserting edges between variables for which the score implies a conditional independence. Building on these ideas, we developed LGES (Alg. 1), which replaces GES' strictly greedy step with a more careful search. We proved that LGES with SAFEINSERT asymptotically recovers the true Markov equivalence class (Thm. 1, Cor. 1), and showed that LGES can incorporate prior knowledge while remaining robust to misspecification in the knowledge. To extend these advancements beyond purely observational settings, we developed $\mathcal{I}$-ORIENT (Alg. 2, Thm. 2), a theoretically sound and scalable score-based method for refining an observational MEC using interventional data. Across experiments on random graphs of varying sizes and densities, our less greedy strategies consistently improved both the accuracy and the efficiency existing GES-style algorithms (Sec. 5). The moral, if there is one, is simple: even in causal discovery, temperance is a virtue.

## Acknowledgements

This research is supported in part by the NSF, ONR, AFOSR, DoE, Amazon, JP Morgan, and The Alfred P. Sloan Foundation. We thank Jonghwan Kim and the anonymous reviewers for their insightful comments.

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

# Appendices

## Contents

## A   Background and related works

### A.1   Definitions and previous results

First, we provide definitions and results used in the main text.

**Definition A.1** ($d$-separation [41])**.** Given a causal DAG $\mathcal{G}$, a node $W$ on a path $\pi$ is said to be a collider on $\pi$ if $W$ has converging arrows into $W$ in $\pi$, e.g., $\rightarrow W \leftarrow$ or $\leftrightarrow W \leftarrow$. $\pi$ is said to be blocked by a set $\mathbf{Z}$ if there exists a node $W$ on $\pi$ satisfying one of the following two conditions: 1) $W$ is a collider, and neither $W$ nor any of its descendants are in $\mathbf{Z}$, or 2) $W$ is not a collider, and $W$ is in $\mathbf{Z}$. Given disjoint sets $\mathbf{X}, \mathbf{Y}$, and $\mathbf{Z}$ in $\mathcal{G}$, $\mathbf{Z}$ is said to *d-separate* $\mathbf{X}$ from $\mathbf{Y}$ in $\mathcal{G}$ if $\mathbf{Z}$ blocks every path from a node in $\mathbf{X}$ to a node in $\mathbf{Y}$ according to the $d$-separation criterion.

**Definition A.2** (DELETE operator, [9, Def. 13])**.** For adjacent nodes $X, Y$ in $\mathcal{E}$ connected as either $X \rightarrow Y$ or $X - Y$, and for any $\mathbf{H} \subseteq \mathbf{Ne}_Y^{\mathcal{E}} \cap \mathbf{Adj}_X^{\mathcal{E}}$, the DELETE$(X, Y, \mathbf{T})$ operator modifies $\mathcal{E}$ by deleting the edge between $X$ and $Y$, and for each $T \in \mathbf{T}$, directing any undirected edges $X - T$ as $X \rightarrow T$ and any $Y - T$ as $Y \rightarrow T$.

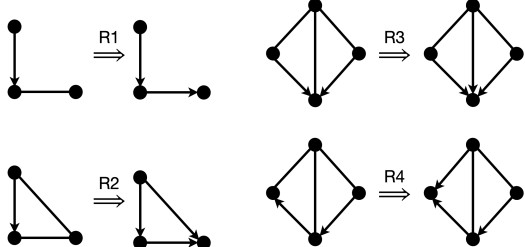

Figure A.1.1: Meek orientation rules for completing partially directed acyclic graphs

The properties and implementation of the TURN operator can be found in [24, Sec. 4.3], though the authors do not provide an exact definition we can reproduce here.

**Definition A.3** (Decomposable scoring criterion [9, Sec. 2.3]). Let $\mathbf{D}$ be a set of data consisting of iid samples from some distribution $P(\mathbf{v})$. A scoring criterion $S$ is said to be *decomposable* if it can be written as a sum of measures, each of which is a function of only a single node and its parents, as

$$S(\mathcal{G}, \mathbf{D}) = \sum_{V_i \in \mathbf{V}} s(v_i, pa_i^{\mathcal{G}})$$

Each local score $s(v_i, pa_i^{\mathcal{G}})$ depends only on the values of $V_i$ and $\mathbf{Pa}_i$ in $\mathbf{D}$.

**Definition A.4** (Consistent scoring criterion [9, Def. 5]). Let $\mathbf{D}$ be a set of data consisting of iid samples from some distribution $P(\mathbf{v})$. A scoring criterion $S$ is said to be *consistent* if, as the number of samples goes to infinity, the following two properties hold for any DAGs $\mathcal{G}, \mathcal{H}$:

1. If $P(\mathbf{v})$ is Markov with respect to $\mathcal{G}$ but not $\mathcal{H}$, then $S(\mathcal{G}, \mathbf{D}) > S(\mathcal{H}, \mathbf{D})$.

2. If $P(\mathbf{v})$ is Markov with respect to both $\mathcal{G}$ and $\mathcal{H}$, but $\mathcal{G}$ contains fewer free parameters than $\mathcal{H}$, then $S(\mathcal{G}, \mathbf{D}) > S(\mathcal{H}, \mathbf{D})$.

**Definition A.5** (Score-equivalent scoring criterion [9, Sec 2.3]). Let $\mathbf{D}$ be a set of data consisting of iid samples from some distribution $P(\mathbf{v})$. A scoring criterion $S$ is said to be *score-equivalent* if, as the number of samples goes to infinity, for any two DAGs $\mathcal{G}, \mathcal{H}$ that are Markov equivalent, $S(\mathcal{G}, \mathbf{D}) = S(H, \mathbf{D})$.

**Definition A.6** (Soft unconditional intervention [24, Sec. 2.1]). A soft unconditional intervention on a set of variables $\mathbf{X}$ sets the value of each variable $V_i \in \mathbf{X}$ to an independent random variable $U_i$ from a given set of random variables $\mathbf{U}$. The resulting distribution is given by

$$P_{\mathbf{X}}(\mathbf{v}) = \prod_{V_i \notin \mathbf{X}} P(v_i \mid pa_i) \prod_{V_i \in \mathbf{X}} P^*(v_i)$$

where $P^*(v_i)$ denotes the distribution of $U_i \in \mathbf{U}$ corresponding to $V_i \in \mathbf{X}$.

**Definition A.7** ($\mathcal{I}$-Markov property [24, Def. 7]). Let $\mathbf{V}$ be a set of variables, $\mathcal{G}$ a causal DAG over $\mathbf{V}$, $\mathcal{I}$ a family of interventional targets, and $(\mathbf{P}_{\mathbf{I}}(\mathbf{v}))_{\mathbf{I} \in \mathcal{I}}$ a corresponding family of interventional distributions. We say $(\mathbf{P}_{\mathbf{I}}(\mathbf{v}))_{\mathbf{I} \in \mathcal{I}}$ satisfies the $\mathcal{I}$-Markov Property of $\mathcal{G}$ if:

1. Each $P_{\mathbf{I}}(\mathbf{v})$ is Markov with respect to the interventional graph $\mathcal{G}_{\bar{\mathbf{I}}}$, and

2. For interventions $\mathbf{I}, \mathbf{J} \in \mathcal{I}$ and variables $V_i \notin \mathbf{I} \cup \mathbf{J}$, $P_{\mathbf{I}}(v_i \mid pa_i) = P_{\mathbf{J}}(v_i \mid pa_i)$.

We let $\mathcal{M}_{\mathcal{I}}(\mathcal{G})$ denote the set of all $(\mathbf{P}_{\mathbf{I}}(\mathbf{v}))_{\mathbf{I} \in \mathcal{I}}$ that are $\mathcal{I}$-Markov with respect to $\mathcal{G}$. Two causal DAGs $\mathcal{G}, \mathcal{H}$ are *$\mathcal{I}$-Markov equivalent* if $\mathcal{M}_{\mathcal{I}}(\mathcal{G}) = \mathcal{M}_{\mathcal{I}}(\mathcal{H})$.

**Meek orientation rules.** In Fig. A.1.1, we provide Meek's orientation rules used in $\mathcal{I}$-ORIENT to orient an $\mathcal{I}$-MEC. These rules provide an algorithm for completing a PDAG to a *completed* PDAG. They are applied repeatedly to a PDAG until no eligible motifs exist.

Next, introduce some additional definitions and results that will be used in Sec. C.

The *skeleton* of a causal DAG $\mathcal{G}$ (denoted $skel(\mathcal{G})$) is the undirected graph that results from ignoring the edge directions of every edge in $\mathcal{G}$. A triplet of variables $(X, Z, Y)$ in $\mathcal{G}$ is said to be *unshielded*

if $(X, Z)$ and $(Y, Z)$ are adjacent but $(X, Y)$ are not. An unshielded triplet is said to be a *v-structure* (or *unshielded collider*) if it is oriented as $X \to Z \leftarrow Y$ in $\mathcal{G}$.

**Theorem A.1** (Graphical criterion for Markov equivalence [57, Thm. 1]). *Two DAGs are Markov equivalent if and only if they have the same skeletons and same v-structures.*

Based on the above characterization, to obtain the CPDAG for the MEC corresponding to a DAG $\mathcal{G}$, one adds an undirected edge for every adjacency in $\mathcal{G}$; orients any v-structures according to $\mathcal{G}$; then applies Meek's orientation rules to complete the resulting PDAG to a CPDAG.

**Definition A.8** (Global Markov property [41]). A probability distribution $P(\mathbf{v})$ over a set of variables $\mathbf{V}$ is said to satisfy the global Markov property for a causal DAG $\mathcal{G}$ if, for arbitrary disjoint sets $\mathbf{X}, \mathbf{Y}, \mathbf{Z} \subset \mathbf{V}$ with $\mathbf{X}, \mathbf{Y} \neq \emptyset$,

$$\mathbf{X} \perp_d \mathbf{Y} | \mathbf{Z} \implies \mathbf{X} \perp\!\!\!\perp \mathbf{Y} | \mathbf{Z} \text{ in } P(\mathbf{v}).$$

Let $\mathbf{Nd}_X^{\mathcal{G}}$ denote the set of non-descendants of a variable $X$ in $\mathcal{G}$, i.e. variables in $\mathcal{G}$ (excluding $X$ itself) to which there is no directed path from $X$.

**Definition A.9** (Local Markov property [41]). A probability distribution $P(\mathbf{v})$ over a set of variables $\mathbf{V}$ is said to satisfy the local Markov property for a causal DAG $\mathcal{G}$ if, for every variable $X \in \mathbf{V}$,

$$X \perp\!\!\!\perp \mathbf{Nd}_X^{\mathcal{G}} \mid \mathbf{Pa}_X^{\mathcal{G}} \text{ in } P(\mathbf{v}).$$

**Proposition A.1** (Equivalence of Local and Global Markov Properties [30, Prop. 4]). *Let $\mathcal{G}$ be a causal DAG over variables $\mathbf{V}$. A probability distribution over $\mathbf{V}$ satisfies the global Markov property for $\mathcal{G}$ if and only if it satisfies the local Markov property for $\mathcal{G}$.*

**Definition A.10** (Covered edge [8, Def. 2]). An edge $X \to Y$ in a given causal DAG $\mathcal{G}$ is said to be *covered* if $\mathbf{Pa}_Y^{\mathcal{G}} = \mathbf{Pa}_X^{\mathcal{G}} \cup \{X\}$.

**Lemma A.1** (Covered edge reversal [8, Lemma. 1]). *Let $\mathcal{G}$ be any causal DAG containing the edge $X \to Y$ and let $\mathcal{G}'$ be the DAG that is identical to $\mathcal{G}$ except it instead contains the edge $Y \to X$. Then, $\mathcal{G}'$ is a DAG that is Markov equivalent to $\mathcal{G}$ iff the edge $X \to Y$ is covered in $\mathcal{G}$.*

**Theorem A.2** (Transformational characterization of Markov equivalent graphs [8, Thm. 2]). *Let $\mathcal{G}, \mathcal{G}'$ be a pair of Markov equivalent causal DAGs. Then, there exists a sequence of covered edge reversals transforming $\mathcal{G}$ to $\mathcal{G}'$.*

**Theorem A.3** (Chickering-Meek theorem [9, Thm. 4]). *let $\mathcal{G}$ and $\mathcal{H}$ be a pair of causal DAGs such every $d$-separation that holds in $\mathcal{H}$ also holds in $\mathcal{G}$. Then, there exists a sequence of edge additions and covered edge reversals transforming $\mathcal{G}$ to $\mathcal{H}$ such that after each reversal and addition, $\mathcal{G}$ is DAG and every $d$-separation that holds $\mathcal{H}$ also holds in $\mathcal{G}$.*

## A.2 Related works

### A.2.1 Learning from observational data

Algorithms for causal discovery fall into three broad categories: constraint-based, score-based, and hybrid. Constraint-based algorithms like PC [54] and the Sparsest Permutations (SP) algorithm [44] learn the true MEC using statistical tests for whether the chosen type of constraints, typically conditional independencies, hold in the data. A challenge to these approaches is improving the accuracy of conditional independence tests, for e.g. in controlling type I error [52]. Score-based algorithms such as GES [9, 34] use a scoring criterion that reflects fit between data and graph, typically in the form of a likelihood plus a complexity penalty. Hybrid algorithms such as max-min hill climbing [56] use a combination of the two approaches, for e.g., first learning the skeleton using a constraint-based method then orienting edges in the skeleton using a score. There is no general claim about the relative accuracy of these methods; we refer readers to [56] for an extensive empirical analysis, who found, for instance, that GES outperforms PC in accuracy across various sample sizes [56, Tables 4, 5] .

Commonly, causal discovery algorithms struggle with scaling in high-dimensional settings. This has motivated variants such as Parallel-PC [31] and Fast Greedy Equivalence Search (FGES) [43], which offer faster, parallelized implementations of the algorithms. While FGES offers an additional heuristic over GES, i.e., not adding any edge $X \to Y$ in the forward phase if $X, Y$ are uncorrelated in the data, this heuristic is not theoretically guaranteed to recover the true MEC. More recent continuous-optimization based approaches such as NoTears [61] are in principle more scalable, but lack theoretical guarantees and show brittle performance even in simulated settings [38, 45].

### A.2.2 Learning with prior knowledge

Causal discovery with background knowledge is a well-studied problem [14]. Such background knowledge may be provided by a domain expert or even a large language model [33].

**Perfect background knowledge.** Many algorithms under this umbrella, such as tiered-FCI [50], the K2 algorithm [15], and Knowledge-Guided GES [22], assume that the background knowledge is correct, i.e., consistent with the ground truth. They use this knowledge to constrain the search, for e.g., by never inserting user-forbidden edges. If the expert's background knowledge is imperfect, such methods necessarily fail to recover the true MEC. Even if the knowledge is perfect, it is unknown whether these methods will output the true graph; for e.g., in Knowledge-Guided GES [22], restricting insertions no longer guarantees reachability of an MEC with respect to which the given data is Markov in the forward phase, as in GES.

**Misspecified background knowledge.** The constraint-based approach in [39] allows some misspecification in the expert knowledge in the form of missing or excess adjacencies but not incorrect orientations. However, their approach does not guarantee recovery of the true MEC. Certain score-based approaches treat knowledge about edges a 'soft' prior [1, 4, 7, 26, 40] to guide the search, but lack theoretical guarantees on the output graph. One exception with theoretical guarantees is the Sparsest Permutations (SP) algorithm [44], which can initialize the search to an ordering over variables provided by an expert.

### A.2.3 Learning from interventional data

Observational learning algorithms can only learn a causal graph up to its observational Markov equivalence class (MEC). The MEC is the limit of what can be identified from observational data without further assumptions. However, MECs can often be large and uninformative for downstream causal tasks [25]. Interventional data can help significantly refine observational MECs [24], and is becoming increasingly available, for e.g., in biological settings due to advances in single-cell technologies [17, 49]. This has motivated the design of algorithms for causal discovery from observational and interventional data such as the score-based Greedy Interventional Equivalence Search (GIES) [24] and the CI-based Interventional Greedy Sparsest Permutations (I-GSP) [58]. However, in [58], it was shown that GIES is inconsistent, i.e.. not guaranteed to recover the true interventional MEC in the sample limit. I-GSP is asymptotically consistent, but being permutation-based, struggles to scale in high-dimensional settings.

## B   Discussion and examples

### B.1   Design of Less Greedy Equivalence Search

In this section, we elaborate on certain design choices made in the LGES algorithm.

**Choice of DAG in SafeInsert.** As specified in Def. 2 and Alg. 5, at a given MEC $\mathcal{E}$, SafeInsert chooses some DAG $\mathcal{G}$ from $\mathcal{E}$ to perform its check for conditional independence. In theory, any any procedure for choosing a DAG from an MEC suffices. In our implementation, we use Alg. 7, a simple polynomial-time algorithm for converting a CPDAG to DAG presented in [18] and used in the original GES [9] Another possibility is the polynomial-time algorithm for uniform random sampling of a DAG from an MEC [59]. We leave investigation of DAG choice to future work.

**Backward phase of LGES.** LGES only modifies the forward phase of GES, while keeping the backward phase intact. Since a score-decreasing INSERT implies independence under some conditioning set (Prop. 1), discarding all $X - Y$ insertions when a single score-decreasing $\text{INSERT}(X, Y, \mathbf{T})$ operator is found promotes more robust edge insertions, i.e., between variables that are not found to be conditionally independent. This significantly reduces false adjacencies in the learned graph (Fig. D.1.1). One may ask, why not use an analogous strategy in the backward phase, discarding all $X - Y$ deletions when a single score-decreasing $\text{DELETE}(X, Y, \mathbf{T})$ operator is found? This is because one score-decreasing deletion implies dependence given one conditioning set, not given any conditioning set. However, a true $X - Y$ adjacency implies dependence given any conditioning set. So, skipping all deletions due to one score decrease would lead to many false adjacencies and

compromise soundness. A true adjacency would imply that all $X - Y$ deletes are score-decreasing, so LGES/GES will not apply any of them. Experiments support this: false non-adjacencies are rare (Fig. D.1.1).

**Contrast with PC algorithm.** There is a surface-level similarity between LGES and the PC algorithm [54]. The PC algorithm begins with a complete graph, then removes edges between variables for which it finds some separating set. In particular, for adjacent variables $X, Y$ in the current graph $\mathcal{G}$, it iterates over all subsets $\mathbf{Z} \subseteq \mathbf{Adj}_X^{\mathcal{G}} \cup \mathbf{Adj}_Y^{\mathcal{G}}$ to find some $\mathbf{Z}$ such that $X \perp\!\!\!\perp Y \mid \mathbf{Z}$. If $X, Y$ are non-adjacent in the ground-truth graph, then the PC algorithm is guaranteed to find a $\mathbf{Z}$ which separates them.

LGES, on the other hand, starts with an empty (or user-provided) graph. It then avoids inserting the edge $X \to Y$ if it finds a set separating $X$ and $Y$. However, LGES only searches over restricted space of separating sets. CONSERVATIVEINSERT checks if $X \perp\!\!\!\perp Y \mid \mathbf{Pa}_Y^{\mathcal{G}}$ for any $\mathcal{G}$ in the current MEC. SAFEINSERT checks if $X \perp\!\!\!\perp Y \mid \mathbf{Pa}_Y^{\mathcal{G}}$ for a fixed $\mathcal{G}$ in the current MEC. There may exist a set $\mathbf{Z}$ separating $X$ and $Y$ that does not equal $\mathbf{Pa}_Y^{\mathcal{G}}$ for some $\mathcal{G}$ in the current MEC. LGES may not find this $\mathbf{Z}$, and thus insert an edge between $X$ and $Y$. In this way, it differs from the PC algorithm.

The advantage of LGES is that it does not require conducting any scoring operations beyond those which GES already conducts. On the other hand, an approach which iterates over all possible separating sets of $X$ and $Y$ would require significantly more scoring operations. Still, future work might consider the theoretical guarantees and empirical performance of such a score-based approach.

## B.2 Limitations of Less Greedy Equivalence Search

**Causal sufficiency.** In this work, we assume that the underlying system is *Markovian*, i.e. no two observed variables have an unobserved common cause. This is also known as the *causal sufficiency* assumption. While this assumption is standard in causal discovery, it can be violated in practice. In settings with unobserved confounders, i.e., *non-Markovian* settings, the equivalence class of graphs that can be identified is typically even larger (and hence more uninformative) than in the Markovian case. One reason for this is that two variables may be non-adjacent in the true graph while still being inseparable by any set due to the existence of an *inducing path* between them [57, Def. 2]. As a result, performing causal inference from equivalence classes of non-Markovian graphs is challenging.

Still, there has been work on causal discovery in non-Markovian settings. Constraint-based approaches include the FCI algorithm [53, 60] and its interventional variants [27, 29, 32], guaranteed to recover the true equivalence class in the sample limit. While there has been progress towards score-based approaches [6, 13, 46, 55], finding algorithms that are asymptotically correct remains an open problem.

There is a fundamental theoretical challenge to generalizing our approach to the non-Markovian setting. GES relies on two useful properties of Markovian DAGs. First is the local Markov property [30] of such DAGs, allowing for a locally consistent scoring criterion (Def. 1). Second are the transformational characterizations of two Markovian causal DAGs $\mathcal{G}_1$ and $\mathcal{G}_2$ when (a) $\mathcal{G}_1$ and $G_2$ encode exactly the same $d$-separations (Thm. A.2) and (b) (a) $\mathcal{G}_1$ encodes a subset of the $d$-separations encoded in $\mathcal{G}_2$ (Thm. A.3). A 'transformational characterization' is a procedure whereby $\mathcal{G}_1$ can be transformed into $\mathcal{G}_2$ by a sequence of single-edge changes that satisfy certain criteria. The transformational characterization of (a) has been generalized to non-Markovian causal DAGS [60]. Moreover, there exists a local Markov property for non-Markovian causal DAGs, as well as an efficient algorithm to test condition (b) [28]. However, it remains an open problem to fully recover (b) in the non-Markovian setting.

**Other assumptions.** In this work, we assume we are given a distribution that is Markov and *faithful* with respect to some causal DAG. This is a standard assumption in causal discovery, often justified by the fact that the set of distributions that are Markov but not faithful with respect to a given DAG has Lebesgue measure zero. Moreover, if we assume only that the given distribution is Markov with respect to some causal DAG, we can never rule out the true DAG being the fully connected DAG. Still, there has been work on relaxing the faithfulness assumption, giving rise to the Sparsest Permutations algorithm [44].

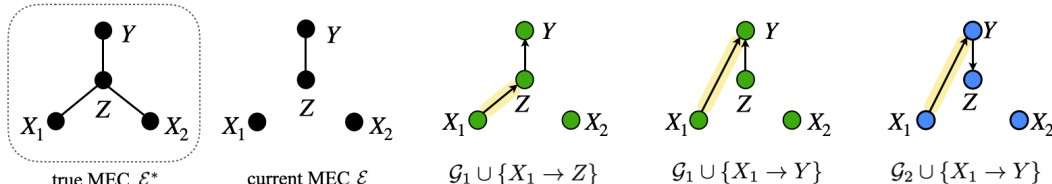

Figure B.3.1: Figs. 2, 3 partially reproduced for convenience. GES is given a distribution $P(\mathbf{v})$ whose true MEC is represented by $\mathcal{E}^*$ (left). GES is currently at MEC $\mathcal{E}$, evaluating which INSERT operator to apply next. Each INSERT corresponds to picking some $\mathcal{G} \in \mathcal{E}$ and adding some edge to it.

We further assume in this work that we are given a scoring criterion that is decomposable and consistent. This does not strictly mean we make parametric assumptions. Existing scores such as BIC satisfy these criteria as long as the model is a *curved exponential family* [9, 23]. This includes multinomial (discrete) and linear-Gaussian models. For continuous data, the linear-Gaussian assumption can be violated in practice. In this case, one can discretize the data before using it for causal discovery, as in [48]. However, since the parameter space of multinomial models is quite large, and information is lost during discretization, it would be valuable future work to investigate scores for other models of continuous data.

### B.3 Extended example of GES trajectories

Finally, we return to our explanation of the two trajectories GES might take in Ex. 2, Fig. 2.

**Example B.1.** (Ex. 2 continued). Recall that GES is given a distribution $P(\mathbf{v})$ whose true MEC is $\mathcal{E}^*$ (Fig. B.3.1, left). GES is currently at the MEC $\mathcal{E}$, evaluating which INSERT operator to apply. Each INSERT operator corresponds to picking some DAG $\mathcal{G} \in \mathcal{E}$ and adding some edge to it. One such operator corresponds to picking $\mathcal{G}_1 \in \mathcal{E}$, and adding the edge $X_1 \to Z$ to it. Another such operator corresponds to picking $\mathcal{G}_2 \in \mathcal{E}$, and adding the edge $X_1 \to Y$ to it. Both operators, shown in Fig. B.3.1, result in a score increase by local consistency (Def. 1) since $Z \not\perp\!\!\!\perp X_1 \mid \mathbf{Pa}_Z^{\mathcal{G}_1}$ and $Y \not\perp\!\!\!\perp X_1 \mid \mathbf{Pa}_Y^{\mathcal{G}_2}$ in $P(\mathbf{v})$. Although $\mathcal{G}_1 \cup X_1 \to Z$ looks 'closer' to the true MEC $\mathcal{E}^*$ than $\mathcal{G}_2 \cup X_! \to Y$, when tested empirically, the latter often scores more than the former (Ex. 2).

Moreover, even in the sample limit, the consistency (Def. A.4) of the scoring criterion does not guarantee that $\mathcal{G}_2 \cup \{X_1 \to Y\}$ will score lower than $\mathcal{G}_1 \cup \{X_1 \to Z\}$. Neither the global nor the local consistency of the score provide an immediate guarantee for which will score higher. Global consistency only allows us to compare graphs when $P(\mathbf{v})$ is Markov with respect to at least one of them; however, $P(\mathbf{v})$ is not Markov with respect to $\mathcal{G}_2 \cup \{X_1 \to Y\}$ or $\mathcal{G}_1 \cup \{X_1 \to Z\}$. Local consistency (Def. 1) does not let us compare these graphs either, since they differ by more than an edge addition. Therefore, GES may move either to the MEC of $\mathcal{G}_1 \cup \{X_1 \to Z\}$ (as in $\tau_1$, Fig. 2) or to the MEC of $\mathcal{G}_2 \cup \{X_1 \to Y\}$ (as in $\tau_2$, Fig. 2). It is unknown a priori which operator is the highest-scoring. □

## C Proofs and pseudocode

In this section, we provide proofs, additional results, and pseudocode for the algorithms of the main paper.

*Theorem* 1 (Correctness of GGES). Let $\mathcal{E}$ denote the Markov equivalence class that results from GGES (Alg. 3) initialized from an arbitrary MEC $\mathcal{E}_0$ and let $P(\mathbf{v})$ denote the distribution from which the data $\mathbf{D}$ was generated. Then, as the number of samples goes to infinity, $\mathcal{E}$ is the Markov equivalence class underlying $P(\mathbf{v})$.

*Proof.* The proof is similar to that of [9, Lemma 9, 10]. First, we will prove the correctness of GGES run with the forward-turning-backward phase order.

Forward phase. First, we show that $P(\mathbf{v})$ is Markov with respect to the MEC $\mathcal{E}'$ resulting from the forward phase of GGES. Let $\mathcal{G}$ be any DAG in $\mathcal{E}'$. By assumption, since GETOPERATOR does not find any valid score-increasing INSERT operators, there exists no such operator. Since there exist no

score-increasing INSERT operators, the local consistency of the scoring criterion (Def. 1) implies that for every $X \in \mathcal{G}$ and $Y \in \mathbf{Nd}_X^{\mathcal{G}}$, $X \perp\!\!\!\perp Y \mid \mathbf{Pa}_X^{\mathcal{G}}$. Otherwise, if we had some $X \in \mathcal{G}$ and $Y \in \mathbf{Nd}_X^{\mathcal{G}}$ such that $X \not\perp\!\!\!\perp Y \mid \mathbf{Pa}_X^{\mathcal{G}}$, the INSERT$(Y, X, *)$ operator corresponding to $\mathcal{G} \cup \{Y \to X\}$ would result in a score increase. Since $P(\mathbf{v})$ is faithful to some DAG, it satisfies the *composition axiom* of conditional independence [41]: if $X \perp\!\!\!\perp Y \mid \mathbf{Pa}_X^{\mathcal{G}}$ for every $Y \in \mathbf{Nd}_X^{\mathcal{G}}$, then $X \perp\!\!\!\perp \mathbf{Nd}_X^{\mathcal{G}} \mid \mathbf{Pa}_X^{\mathcal{G}}$. Since this is true for every $X$, $P(\mathbf{v})$ satisfies the local Markov property of $\mathcal{G}$. By the equivalence of the global and local Markov properties (Prop. A.1), this means every d-separation in $\mathcal{G}$ implies a corresponding CI in $P(\mathbf{v})$. Thus, $P(\mathbf{v})$ is Markov with respect to $\mathcal{G}$ and hence to $\mathcal{E}'$.

Turning phase. Next, we show that $P(\mathbf{v})$ is Markov with respect to the MEC $\mathcal{E}''$ resulting from the turning phase of GGES. The turning phase starts with $\mathcal{E}'$, output by the forward phase. We have shown that $P(\mathbf{v})$ is Markov with respect to $\mathcal{E}'$. By construction, each TURN operator applied to the current MEC in the backward phase results in a score increase If any operator resulted in an $\mathcal{E}''$ with respect to which $P(\mathbf{v})$ is not Markov, the consistency of the scoring criterion A.4 implies that it would decrease the score. Therefore, $P(\mathbf{v})$ must be Markov with respect to $\mathcal{E}''$.

Backward phase. Finally, we show that $P(\mathbf{v})$ is both Markov and faithful to the MEC $\mathcal{E}$ resulting from the backward phase of GGES. The argument that $P(\mathbf{v})$ is Markov with respect to $\mathcal{E}$ is similar to that of the turning phase above. It remains to show that $P(\mathbf{v})$ is faithful to $\mathcal{E}$. Let $\mathcal{E}^*$ be the true MEC underlying $P(\mathbf{v})$. Since $P(\mathbf{v})$ is both Markov and faithful with respect to $\mathcal{E}^*$, and $P(\mathbf{v})$ is Markov with respect to $\mathcal{E}$, every $d$-separation in $\mathcal{E}$ must also hold in $\mathcal{E}^*$. Then, by the Chickering-Meek theorem (Thm. A.3), for any $\mathcal{G} \in \mathcal{E}$ and $\mathcal{H} \in \mathcal{E}^*$, there exists a sequence of covered edge reversals and edge additions that transform $\mathcal{H}$ to $\mathcal{G}$. If this sequence only contains covered edge reversals, then $\mathcal{H}$ and $\mathcal{G}$ are Markov-equivalent (Lemma. A.1), and we are done. Otherwise, let the last edge addition in this sequence add the edge $X \to Y$, resulting in the DAG $\mathcal{G}'$. Since $\mathcal{G}'$ can be transformed to $\mathcal{G}$ by a sequence of covered edge reversals, they are Markov equivalent, and we have $\mathcal{G}' \in \mathcal{E}$ (Lemma. A.1). Moreover, since this sequence of transformations includes only covered edge reversals and edge additions, and $P(\mathbf{v})$ is Markov with respect to $\mathcal{H}$, $P(\mathbf{v})$ is also Markov with respect to $\mathcal{G}' \setminus \{X \to Y\}$ (by Lemma. A.1, covered edge reversals and additions do not create additional $d$-separations). By the consistency of the scoring criterion, $\mathcal{G}' \setminus \{X \to Y\}$ has a higher score than $\mathcal{G}' \in \mathcal{E}$ since the former has fewer parameters. The corresponding DELETE operator thus results in a score increase, and by assumption, GETOPERATOR is guaranteed to find some score-increasing deletion in this case. Thus, we have a contradiction.

Next, we will prove the correctness of GGES without the phase structure, i.e., when GETOPERATOR returns a score-increasing INSERT or DELETE operator if one exists. The proof is similar to the previous case. GGES terminates at an MEC $\mathcal{E}$ only when GETOPERATOR finds no score-increasing INSERT *or* DELETE operator. By assumption, this implies no such operator exists. If no score-increasing increasing INSERT operator exists, by the same argument as for the forward phase above, the given $P(\mathbf{v})$ must be Markov with respect to $\mathcal{E}$. Given that $P(\mathbf{v})$ is Markov with respect to $\mathcal{E}$, and, furthermore, no score-increasing increasing DELETE operator exists, then by the same argument as for the backward phase above, the given $P(\mathbf{v})$ must be both Markov and faithful to $\mathcal{E}$. $\qquad\square$

*Proposition* 1. Let $\mathcal{E}$ denote an arbitrary CPDAG and let $P(\mathbf{v})$ denote the distribution from which the data $\mathbf{D}$ was generated. Assume, as the number of samples goes to infinity, that there exists a valid score-decreasing INSERT$(X, Y, \mathbf{T})$ operator for $\mathcal{E}$. Then, there exists a DAG $\mathcal{G} \in \mathcal{E}$ such that (1) $Y \perp_d X \mid \mathbf{Pa}_Y^{\mathcal{G}}$ and (2) $Y \perp\!\!\!\perp X \mid \mathbf{Pa}_Y^{\mathcal{G}}$ in $P(\mathbf{v})$.

*Proof.* The score change of a valid INSERT$(X, Y, \mathbf{T})$ corresponds to picking a DAG $\mathcal{G} \in \mathcal{E}$ and comparing $S(\mathcal{G}, \mathbf{D})$ with $S(\mathcal{G} \cup \{X \to Y\}, \mathbf{D})$. By local consistency (Def. 1), if $Y \not\perp\!\!\!\perp X \mid \mathbf{Pa}_Y^{\mathcal{G}}$, then the score must increase. By contrapositive, we have $Y \perp\!\!\!\perp X \mid \mathbf{Pa}_Y^{\mathcal{G}}$ in $P(\mathbf{v})$. Moreover, since $X$ must be a non-descendant of $Y$ for $\mathcal{G} \cup \{X \to Y\}$ to be a DAG, $Y \perp_d X \mid \mathbf{Pa}_Y^{\mathcal{G}}$ in $\mathcal{G}$. $\qquad\square$

We provide the following guarantee for LGES Alg. 1 run with CONSERVATIVEINSERT (Strategy 1).

**Proposition C.1** (Partial guarantee on CONSERVATIVEINSERT). *Let LGES* * *denote the variant of LGES (Alg. 1) that uses* CONSERVATIVEINSERT *instead of* SAFEINSERT *in the forward phase. Let* $\mathcal{E}$ *denote the equivalence class that results from the forward phase of LGES* * *initialized to an arbitrary MEC* $\mathcal{E}_0$, *let* $P(\mathbf{v})$ *denote the distribution from which the data* $\mathbf{D}$ *was generated, and let* $\mathcal{E}^*$ *be the true MEC underlying* $P(\mathbf{v})$. *Then, as the number of samples goes to infinity,*

1. $skel(\mathcal{E}^*) \subseteq skel(\mathcal{E})$

2. *For any unshielded triplet $(X, Z, Y) \in \mathcal{E}^*$, either $X, Y$ are adjacent in $\mathcal{E}$ or $(X, Z, Y)$ is a collider in $\mathcal{E}^*$ if and only if it is a collider in $\mathcal{E}$.*

*Proof.* For any variables $X, Y$ adjacent in $\mathcal{E}^*$, since $P(\mathbf{v})$ is faithful to $\mathcal{E}^*$, $X, Y$ are not independent in the data conditional on any set $\mathbf{Z} \subseteq \mathbf{V}$. Hence, for any $\mathcal{G} \in \mathcal{E}$, $X \not\perp\!\!\!\perp Y \mid \mathbf{Pa}_Y^{\mathcal{G}}$. Therefore, we always have $s(\mathcal{G}) < s(\mathcal{G} \cup \{X \to Y\})$ for any $\mathcal{G} \in \mathcal{E}$ such that $X \in \mathbf{Nd}_Y^{\mathcal{G}}$, and all valid INSERT$(X, Y, *)$ operators will result in a score increase. Hence, CONSERVATIVEINSERT will consider all such operators. Since $\hat{\mathcal{E}}$ is a local optimum of the score, any variables that are adjacent in $\mathcal{E}^*$ must also be adjacent in $\mathcal{E}$. Therefore, $skel(\mathcal{E}^*) \subseteq skel(\mathcal{E})$.

Then, consider some unshielded triplet $(X, Z, Y) \in \mathcal{E}^*$. Since $skel(\mathcal{E}^*) \subseteq skel(\mathcal{E})$, $(X, Z)$ and $(Y, Z)$ must be adjacent in $\mathcal{E}$. If $(X, Y)$ are also adjacent in $\mathcal{E}$, we are done. Otherwise, we have an unshielded triplet $(X, Z, Y) \in \mathcal{E}$. Assume $(X, Z, Y)$ is a collider in $\mathcal{E}^*$. Since CONSERVATIVEINSERT finds no score-increasing INSERT operators for $\mathcal{E}$, and $X, Y$ are non-adjacent in $\mathcal{E}$, it must be the case that $\exists \mathcal{G} \in \mathcal{E}$ such that $Y \in \mathbf{Nd}_X^{\mathcal{G}}$ and $X \perp\!\!\!\perp Y \mid \mathbf{Pa}_X^{\mathcal{G}}$ or $X \in \mathbf{Nd}_Y^{\mathcal{G}}$ and $X \perp\!\!\!\perp Y \mid \mathbf{Pa}_Y^{\mathcal{G}}$. Without loss of generality, assume it is the former. Since $(X, Z, Y)$ is a collider in $\mathcal{E}^*$, $X \not\perp\!\!\!\perp Y \mid \mathbf{Z}$ in $P(\mathbf{v})$ for any set $\mathbf{Z}$ containing $Z$. Therefore, it must be the case that $Z \notin \mathbf{Pa}_X^{\mathcal{G}}$. Hence, $\mathcal{G}$ contains the edge $X \to Z$. Since $Y \in \mathbf{Nd}_X^{\mathcal{G}}$, this further implies that $\mathcal{G}$ contains the edge $Y \to Z$. Therefore, $(X, Z, Y)$ is a collider in $\mathcal{G}$ and hence $\mathcal{E}^*$. Next, assume that $(X, Z, Y)$ is a collider in $\mathcal{E}$. Then, $Z \notin \mathbf{Pa}_X^{\mathcal{G}}$ and $Z \notin \mathbf{Pa}_Y^{\mathcal{G}}$ for all $\mathcal{G} \in \mathcal{E}$. As before, since CONSERVATIVEINSERT finds no score-increasing INSERT operators for $\mathcal{E}$, and $X, Y$ are non-adjacent in $\mathcal{E}$, it must be the case that $\exists \mathcal{G} \in \mathcal{E}$ such that $Y \in \mathbf{Nd}_X^{\mathcal{G}}$ and $X \perp\!\!\!\perp Y \mid \mathbf{Pa}_X^{\mathcal{G}}$ or $X \in \mathbf{Nd}_Y^{\mathcal{G}}$ and $X \perp\!\!\!\perp Y \mid \mathbf{Pa}_Y^{\mathcal{G}}$. Then, conditioning on $Z$ is not needed to separate $X, Y$ in $P(\mathbf{v})$, which implies that $(X, Z, Y)$ is also a collider in $\mathcal{E}^*$.

$\square$

We give the following condition, sufficient to guarantee that CONSERVATIVEINSERT returns a score-increasing INSERT operator when one exists.

**Proposition C.2** (Conditional guarantee on CONSERVATIVEINSERT)**.** *Let $\mathcal{E}$ denote a Markov equivalence class and let $P(\mathbf{v})$ denote the distribution from which the data $\mathbf{D}$ was generated.*

*Assume the following holds.*

> **Assumption.** *Let $\mathcal{G}, \mathcal{H}$ be two DAGs such that some $d$-separation encoded in $\mathcal{G}$ does not hold in $\mathcal{H}$. Then, there exists a pair of variables $X, Y$ non-adjacent in $\mathcal{G}$ with $Y \in \mathbf{Nd}_X^{\mathcal{G}}$ such that for every $\mathcal{G}'$ Markov-equivalent to $\mathcal{G}$ with $Y \in \mathbf{Nd}_X^{\mathcal{G}'}$, $X \not\perp\!\!\!\perp_d Y \mid \mathbf{Pa}_X^{\mathcal{G}'}$ in $\mathcal{H}$.*

*Then, as the number of samples goes to infinity, CONSERVATIVEINSERT returns a valid score-increasing INSERT operator if and only if one exists.*

*Proof.* Let $\mathcal{E}^*$ indicate the true MEC underlying $P(\mathbf{v})$. If there exists a valid score-increasing INSERT operator for the current state $\mathcal{E}$, then $P(\mathbf{v})$ is not Markov with respect to $\mathcal{E}$. Since $P(\mathbf{v})$ is faithful to $\mathcal{E}^*$, this implies that there exists some $d$-separation encoded in $\mathcal{E}$ that does not hold in $\mathcal{E}^*$. By the assumption, this implies that there exists $X, Y$ non-adjacent in $\mathcal{E}$ such that for every $\mathcal{G}$ in $\mathcal{E}$ with $Y \in \mathbf{Nd}_X^{\mathcal{G}}$, $X \not\perp\!\!\!\perp_d Y \mid \mathbf{Pa}_X^{\mathcal{G}}$ in $\mathcal{E}^*$ and hence $X \not\perp\!\!\!\perp Y \mid \mathbf{Pa}_X^{\mathcal{G}}$ in $P(\mathbf{v})$. Therefore, every INSERT$(Y, X, *$ operator results in a score increase for $\mathcal{E}$. Then, CONSERVATIVEINSERT is guaranteed to find a score-increasing INSERT. The reverse direction follows by construction, since CONSERVATIVEINSERT enumerates only valid INSERT operators and returns one only if it increases the score. $\square$

As a corollary of the above and Thm. 1, we can also show that LGES with CONSERVATIVEINSERT instead of SAFEINSERT is guaranteed to recover the true MEC in the sample limit, if the assumption in Prop. C.2 holds. We leave the correctness of this assumption open.

*Proposition* 2 (Correctness of SAFEINSERT). Let $\mathcal{E}$ denote a Markov equivalence class and let $P(\mathbf{v})$ denote the distribution from which the data $\mathbf{D}$ was generated. Then, as the number of samples goes to infinity, SAFEINSERT returns a valid score-increasing INSERT operator if and only if one exists.

*Proof.* Assume there exists a valid score-increasing INSERT operator for the given MEC $\mathcal{E}$. Then, $P(\mathbf{v})$ is not Markov with respect to $\mathcal{E}$. Hence, $P(\mathbf{v})$ is not Markov with respect to the $\mathcal{G} \in \mathcal{E}$ chosen by SAFEINSERT. By the equivalence of the global and local Markov properties (Prop. A.1), this implies that there exists $X \in \mathcal{G}$ such that $X \not\perp\!\!\!\perp \mathbf{Nd}_X^{\mathcal{G}} \mid \mathbf{Pa}_X^{\mathcal{G}}$. Since $P(\mathbf{v})$ is faithful to some DAG, it satisfies the *composition* axiom of conditional independence [41]; hence, there exists some $Y \in \mathbf{Nd}_X^{\mathcal{G}}$ such that $X \not\perp\!\!\!\perp Y \mid \mathbf{Pa}_X^{\mathcal{G}}$. By the local consistency and decomposability of the scoring criterion, we have $s(X, \mathbf{Pa}_X^{\mathcal{G}}) < s(X, \mathbf{Pa}_X^{\mathcal{G}} \cup \{Y\})$. Then, SAFEINSERT will find the valid score-increasing INSERT$(Y, X, \mathbf{T})$ operator corresponding to $\mathcal{G} \cup \{Y \rightarrow X\}$. The reverse direction is similar. If SAFEINSERT outputs some $(X, Y, \mathbf{T})$, this implies it has found some $X \in \mathcal{G}$ and $Y \in \mathbf{Nd}_X^{\mathcal{G}}$ such $s(X, \mathbf{Pa}_X^{\mathcal{G}}) < s(X, \mathbf{Pa}_X^{\mathcal{G}} \cup \{Y\})$, and hence $X \not\perp\!\!\!\perp Y \mid \mathbf{Pa}_X^{\mathcal{G}}$. This implies that $P(\mathbf{v})$ is not Markov with respect to $\mathcal{G}$ and hence to $\mathcal{E}$, and there exists a valid score-increasing INSERT for $\mathcal{E}$. The INSERT output by SAFEINSERT is a valid score-increasing operator by construction. $\square$

We provide pseudocode for the GETSAFEINSERT procedure in Alg. 5. GETSAFEINSERT generalizes SAFEINSERT; instead of searching for a valid INSERT across all non-adjacencies in $\mathcal{E}$, it searches for a valid INSERT in a subset of the non-adjacencies in $\mathcal{E}$, given by the *candidates* set. This enables the use of the prioritisation scheme of GETPRIORITYINSERTS.

**Proposition C.3** (Correctness of GETPRIORITYINSERTS). *Let priorityList be the list of sets of edges output by* GETPRIORITYINSERTS *(Alg. 6) given a Markov equivalence class $\mathcal{E}$ and prior assumptions $\mathbf{S} = \langle \mathbf{R}, \mathbf{F} \rangle$. Then, the union of all sets of edges in priorityList is equal to the set of variable pairs $(X, Y)$ that are non-adjacent in $\mathcal{E}$.*

*Proof.* This follows from the fact that GETPRIORITYINSERTS loops over all non-adjacencies in $\mathcal{E}$, and any adjacencies not determined by $\mathbf{S}$ are added to $priorityList[3]$ on line 10. $\square$

**Corollary C.1** (Correctness of LGES-0). *Let $\mathcal{E}$ denote the Markov equivalence class that results from LGES-0 (Alg. 8) with* SAFEINSERT, *initialised from an arbitrary MEC $\mathcal{E}_0$ and given prior assumptions $\mathbf{S} = \langle \mathbf{R}, \mathbf{F} \rangle$ using* SAFEINSERT, *and let $P(\mathbf{v})$ denote the distribution from which the data $\mathbf{D}$ was generated. Then, as the number of samples goes to infinity, $\mathcal{E}$ is the Markov equivalence class underlying $P(\mathbf{v})$.*

*Proof.* This follows from Prop. 2, Prop. C.3, and Thm. 1. In the forward phase, if $P(\mathbf{v})$ is not Markov with respect to the current MEC $\mathcal{E}$, SAFEINSERT will find some score-increasing INSERT$(X, Y, \mathbf{T})$(Prop. 2). Since $(X, Y)$ must be in some set in the priority list returned by GETPRIORITYINSERT (Prop. C.3), some call to GETSAFEINSERT will find a score-increasing INSERT operator. Therefore, each forward step is guaranteed to find a valid score-increasing INSERT operator, if it exists. In the backward phase, since LGES-0 enumerates all valid DELETE operators at each step, it is also guaranteed to find a valid score-increasing DELETE operator when if one exists. Thus, LGES-0 terminates only when there are no score-increasing operators for the current MEC. As such, LGES-0 satisfies the conditions of GGES (Alg. 3) and its correctness follows from Thm. 1. $\square$

*Corollary* 1 (Correctness of LGES). Let $\mathcal{E}$ denote the Markov equivalence class that results from LGES (Alg. 1) with SAFEINSERT, initialised from an arbitrary MEC $\mathcal{E}_0$ and given prior assumptions $\mathbf{S} = \langle \mathbf{R}, \mathbf{F} \rangle$, and let $P(\mathbf{v})$ denote the distribution from which the data $\mathbf{D}$ was generated. Then, as the number of samples goes to infinity, $\mathcal{E}$ is the Markov equivalence class underlying $P(\mathbf{v})$.

*Proof.* This follows from Prop. 2, Prop. C.3, and Thm. 1. When LGES terminates, there exists no score-increasing deletion for the current MEC $\mathcal{E}$ by construction. By a similar argument to Cor. C.1, we can further show that there exists no score-increasing insertion for the current MEC $\mathcal{E}$. Thus, LGES satisfies the conditions of GGES (Alg. 3) and its correctness follows from Thm. 1. $\square$

**Corollary C.2** (Correctness of LGES+). *Let $\mathcal{E}$ denote the Markov equivalence class that results from LGES+ (Alg. 8) with* SAFEINSERT, *initialised from an arbitrary MEC $\mathcal{E}_0$ and given prior assumptions $\mathbf{S} = \langle \mathbf{R}, \mathbf{F} \rangle$ using* SAFEINSERT, *and let $P(\mathbf{v})$ denote the distribution from which the*

---

**Algorithm 3:** Generalized Greedy Equivalence Search (GGES)

---
**Input:** Data $\mathbf{D} \sim \mathbf{P}(\mathbf{v})$, initial MEC $\mathcal{E}$, scoring criterion $S$, initial MEC $\mathcal{E}_0$, list of phases
  $phases$ in {['forward', 'backward'], ['any']}
**Output:** MEC $\mathcal{E}$ of $\mathbf{P}(\mathbf{v})$

1 $\mathcal{E} \leftarrow \mathcal{E}_0$;
2 **foreach** $phase$ in $phases$ **do**
3      **repeat**
4          OPERATE$(X, Y, \mathbf{T}) \leftarrow$ GETOPERATOR$(\mathcal{E}, \mathbf{D}, S, phase)$ ;
5          $\mathcal{E} \leftarrow \mathcal{E} +$ OPERATE$(X, Y, \mathbf{T})$ ;
6      **until** *no score-increasing operators exist*;
7 **return** $\mathcal{E}$

---

*data $\mathbf{D}$ was generated. Then, as the number of samples goes to infinity, $\mathcal{E}$ is the Markov equivalence class underlying $P(\mathbf{v})$.*

*Proof.* The correctness of LGES (Cor. 1) implies that the MEC obtained on line 1 of LGES+ is the highest-scoring MEC. Therefore, the condition on line 7 of LGES+ will never be true, and LGES+ outputs the same MEC as LGES. $\qquad\square$

*Theorem* 2 (Correctness of $\mathcal{I}$-ORIENT). Let $\mathcal{E}$ denote the Markov equivalence class that results from $\mathcal{I}$-ORIENT (Alg. 2) given an observational MEC $\mathcal{E}_0$ and interventional targets $\mathcal{I}$, and let $(\mathbf{P_I}(\mathbf{v}))_{\mathbf{I} \in \mathcal{I}}$ denote the family of distributions from which the data $(\mathbf{D_I})_{\mathbf{I} \in \mathcal{I}}$ was generated. Assume that $\mathcal{E}_0$ is the MEC underlying $P_\emptyset(\mathbf{v})$. Then, as the number of samples goes to infinity for each $\mathbf{I} \in \mathcal{I}$, $\mathcal{E}$ is the $\mathcal{I}$-Markov equivalence class underlying $(\mathbf{P_I}(\mathbf{v}))_{\mathbf{I} \in \mathcal{I}}$.

*Proof.* Let $\mathcal{E}^*$ denote the true $\mathcal{I}$-MEC underlying $(\mathbf{P_I}(\mathbf{v}))_{\mathbf{I} \in \mathcal{I}}$. Since $\mathcal{E}$ only orients undirected edges in $\mathcal{E}_0$, and $\mathcal{E}_0$ has the same skeleton and v-structures as $\mathcal{E}^*$, $\mathcal{E}$ also has the same skeleton and v-structures as $\mathcal{E}^*$. Next, we show that for every variable pair $(X, Y)$ adjacent in $\mathcal{E}^*$ (and hence $\mathcal{E}$) for which there exists some $\mathbf{I} \in \mathcal{I}$ with $X \in \mathbf{I}, Y \notin \mathbf{I}$, this edge is directed in both $\mathcal{E}$ and $\mathcal{E}^*$, and moreover, has the same direction in both.

Consider some edge $(X, Y) \in \mathcal{E}^*$ for which there exists $\mathbf{I} \in \mathcal{I}$ such that $X \in \mathbf{I}, Y \notin \mathbf{I}$. Then, $(X, Y)$ is directed and $\mathcal{I}$-essential in $\mathcal{E}$ [24, Cor. 13]. We will show that $\mathcal{E}^*$ contains $Y \to X$ if and only if for every such $\mathbf{I}$, $s_{\mathbf{D_I}}(y) > s_{\mathbf{D_I}}(y, x)$.

$\implies$ Assume $\mathcal{E}^*$ contains $Y \to X$. If $X \to Y \in \mathcal{E}^*$, then $X \to Y$ in every DAG $\mathcal{G} \in \mathcal{E}^*$. This implies that in every $\mathcal{G} \in \mathcal{E}^*$, there are no directed paths from $Y \to X$ in $\mathcal{G}$ and hence $\mathcal{G}_{\bar{\mathbf{I}}}$. Moreover, since all edges into $X$ are removed in $\mathcal{G}_{\bar{\mathbf{I}}}$, there are no directed paths from $X \to Y$ in $\mathcal{G}_{\bar{\mathbf{I}}}$. Since $P_{\mathbf{I}}(\mathbf{v})$ is Markov with respect to $\mathcal{G}_{\bar{\mathbf{I}}}$, this implies $X \perp\!\!\!\perp Y$ in $P_{\mathbf{I}}(\mathbf{v})$. Let $\mathcal{H}$ denote the empty graph over variables $\mathbf{V}$. Since $X \perp\!\!\!\perp Y$ in $P_{\mathbf{I}}(\mathbf{v})$, by the local consistency of the scoring criterion, $\mathcal{H}$ has a higher score than $\mathcal{H} \cup \{X \to Y\}$. By the decomposability of the scoring criterion, this implies $s_{\mathbf{D_I}}(y) > s_{\mathbf{D_I}}(y, x)$. Since $\mathbf{I}$ was arbitrary, this must be true for each $\mathbf{I} \in \mathcal{I}$ such that $X \in \mathbf{I}, Y \notin \mathbf{I}$.

$\impliedby$ Assume that $s_{\mathbf{D_I}}(y) > s_{\mathbf{D_I}}(y, x)$ for some $\mathbf{I} \in \mathcal{I}$. Let $\mathcal{H}$ denote the empty graph over variables $\mathbf{V}$. Then, since $s_{\mathbf{D_I}}(y) > s_{\mathbf{D_I}}(y, x)$, the decomposability of the scoring criterion implies that $\mathcal{H}$ has a higher score than $\mathcal{H} \cup \{X \to Y\}$. If $Y \not\!\perp\!\!\!\perp X$ in $P_{\mathbf{I}}(\mathbf{v})$, the local consistency of the scoring criterion would imply that $\mathcal{H}$ has a lower score than $\mathcal{H} \cup \{X \to Y\}$. By contrapositive, it must be true that $Y \perp\!\!\!\perp X$ in $P_{\mathbf{I}}(\mathbf{v})$. Since $P_{\mathbf{I}}(\mathbf{v})$ is faithful to $\mathcal{G}_{\bar{\mathbf{I}}}$ for some $\mathcal{G} \in \mathcal{E}^*$, this must imply that $X, Y$ are non-adjacent in $\mathcal{G}_{\bar{\mathbf{I}}}$. Since $X, Y$ are adjacent in $\mathcal{E}^*$, and $X \in \mathbf{I}, Y \notin \mathbf{I}$, this implies that $\mathcal{G}$ and $\mathcal{E}^*$ contain $Y \to X$. This further implies that if the supposition is true for some $\mathbf{I} \in \mathcal{I}$ with $X \in \mathbf{I}, Y \notin \mathbf{I}$, it must be true for all of them.

The argument to show that $\mathcal{E}^*$ contains $X \to Y$ if and only if for every $\mathbf{I}$ with $X \in \mathbf{I}, Y \notin \mathbf{I}$, $s_{\mathbf{D_I}}(y) < s_{\mathbf{D_I}}(y, x)$ is analogous.

Moreover, since these statements are true for each $\mathbf{I} \in \mathcal{I}$, they are also true when comparing the sum over sum over all such $\mathbf{I}$: i.e., $\sum_{\mathbf{I} \in \mathcal{I}, X \in \mathbf{I}, Y \notin \mathbf{I}} s_{\mathbf{D_I}}(y)$ vs $\sum_{\mathbf{I} \in \mathcal{I}, X \in \mathbf{I}, Y \notin \mathbf{I}} s_{\mathbf{D_I}}(y, x)$.

Any edge that is directed in $\mathcal{E}$ is either (a) already directed in $\mathcal{E}_0$, in which case it is similarly directed in $\mathcal{E}^*$, (b) oriented on lines 4 or 7 of $\mathcal{I}$-ORIENT, in which case it is similarly directed in $\mathcal{E}^*$ by the above argument, or (c) oriented by the Meek rules on lines 5 or 8, in which case it is a consequence of edges directed due to (a) and (b), in which case it is also similarly directed in $\mathcal{E}^*$. Moreover, the edges directed in $\mathcal{E}^*$ are also due to (a) their being directed in $\mathcal{E}_0$, (b) there existing some $\mathbf{I} \in \mathcal{I}$ which contains exactly one endpoint of that edge, or (c) their being a consequence by the Meek rules of these two edge types. Therefore, edges directed in $\mathcal{E}^*$ are also similarly directed in $\mathcal{E}$, since $\mathcal{I}$-ORIENT directs each such edge type. We thus have that $\mathcal{E} = \mathcal{E}^*$. $\qquad\square$

---

**Algorithm 4:** GETCONSERVATIVEINSERT

**Input:** MEC $\mathcal{E}$, DAG $\mathcal{G} \in \mathcal{E}$, data $\mathbf{D} \sim P(\mathbf{v})$, edge insertion candidates $candidates$, scoring criterion $S$

**Output:** A valid score-increasing INSERT operator for $\mathcal{E}$ from the adjacencies in $candidates$, or $\emptyset$ if none is found.

1   $\Delta S_{max} \leftarrow -\infty$;
2   $(X_{max}, Y_{max}, \mathbf{T}_{max}) \leftarrow \emptyset$;
3   **foreach** $(X, Y)$ *in candidates* **do**
4      **if** $X \in \mathbf{Nd}_Y^{\mathcal{G}}$ *and* $s(Y, \mathbf{Pa}_Y^{\mathcal{G}}) < s(Y, \mathbf{Pa}_Y^{\mathcal{G}} \cup \{X\})$ **then**
5         $\Delta S_{xy} \leftarrow -\infty$;
6         $(\hat{X}, \hat{Y}, \hat{\mathbf{T}}) \leftarrow \emptyset$;
7         **foreach** *valid* $\mathbf{T} \subseteq \mathbf{Ne}_Y^{\mathcal{E}} \setminus \mathbf{Adj}_X^{\mathcal{E}}$ **do**
8            $\Delta S \leftarrow s\big(Y, (\mathbf{Ne}_Y^{\mathcal{E}} \cap \mathbf{Adj}_X^{\mathcal{E}}) \cup \mathbf{T} \cup \mathbf{Pa}_Y^{\mathcal{E}} \cup \{X\}\big) - s\big(Y, (\mathbf{Ne}_Y^{\mathcal{E}} \cap \mathbf{Adj}_X^{\mathcal{E}}) \cup \mathbf{T} \cup \mathbf{Pa}_{Y,}^{\mathcal{E}}\big)$;
9            **if** $\Delta S > \Delta S_{xy}$ **then**
10              $\Delta S_{xy} \leftarrow \Delta S$;
11              $(\hat{X}, \hat{Y}, \hat{\mathbf{T}}) \leftarrow (X, Y, \mathbf{T})$;
12            **else if** $\Delta S < 0$ **then**
13              $\Delta S_{xy} \leftarrow -\infty$;
14              break;
15         **if** $\Delta S_{xy} > \Delta S_{max}$ **then**
16            $\Delta S_{max} \leftarrow \Delta S_{xy}$;
17            $(X_{max}, Y_{max}, \mathbf{T}_{max}) \leftarrow (\hat{X}, \hat{Y}, \hat{\mathbf{T}})$;
18   **return** $(X_{max}, Y_{max}, \mathbf{T}_{max})$

---

**Algorithm 5:** GETSAFEINSERT

**Input:** MEC $\mathcal{E}$, DAG $\mathcal{G} \in \mathcal{E}$, data $\mathbf{D} \sim P(\mathbf{v})$, edge insertion candidates $candidates$, scoring criterion $S$

**Output:** A valid score-increasing INSERT operator for $\mathcal{E}$ from the adjacencies in $candidates$, or $\emptyset$ if none is found.

1   $\Delta S_{max} \leftarrow -\infty$;
2   $(X_{max}, Y_{max}, \mathbf{T}_{max}) \leftarrow \emptyset$;
3   **foreach** $(X, Y)$ *in candidates* **do**
4      **if** $X \in \mathbf{Nd}_Y^{\mathcal{G}}$ *and* $s(Y, \mathbf{Pa}_Y^{\mathcal{G}}) < s(Y, \mathbf{Pa}_Y^{\mathcal{G}} \cup \{X\})$ **then**
5         **foreach** *valid* $\mathbf{T} \subseteq \mathbf{Ne}_Y^{\mathcal{E}} \setminus \mathbf{Adj}_X^{\mathcal{E}}$ **do**
6            $\Delta S \leftarrow s\big(Y, (\mathbf{Ne}_Y^{\mathcal{E}} \cap \mathbf{Adj}_X^{\mathcal{E}}) \cup \mathbf{T} \cup \mathbf{Pa}_Y^{\mathcal{E}} \cup \{X\}\big) - s\big(Y, (\mathbf{Ne}_Y^{\mathcal{E}} \cap \mathbf{Adj}_X^{\mathcal{E}}) \cup \mathbf{T} \cup \mathbf{Pa}_{Y,}^{\mathcal{E}}\big)$;
7            **if** $\Delta S > \Delta S_{max}$ **then**
8              $\Delta S_{max} \leftarrow \Delta S$;
9              $(X_{max}, Y_{max}, \mathbf{T}_{max}) \leftarrow (X, Y, \mathbf{T})$;
10   **return** $(X_{max}, Y_{max}, \mathbf{T}_{max})$

---

**Algorithm 6:** GETPRIORITYINSERTS

---

**Input:** MEC $\mathcal{E}$, prior assumptions $\mathbf{S} = \langle \mathbf{R}, \mathbf{F} \rangle$

1   $priorityList \leftarrow [\{\} \times 4]$;
2   **foreach** $(X, Y)$ *non-adjacent in* $\mathcal{E}$ **do**
3     **if** $X - *Y \in \mathbf{R}$ **then**
4       |   Add $(X, Y)$ to $priorityList[1]$ ;                               `// required`
5     **else if** $Y \to X \in \mathbf{R}$ **then**
6       |   Add $(X, Y)$ to $priorityList[2]$ ;                       `// weakly required`
7     **else if** $X \to Y \in \mathbf{F}$ *or* $X - Y \in \mathbf{F}$ **then**
8       |   Add $(X, Y)$ to $priorityList[4]$ ;                            `// forbidden`
9     **else**
10      |   Add $(X, Y)$ to $priorityList[3]$ ;                            `// ambivalent`
11   **return** $priorityList$

---

---

**Algorithm 7:** PDAGTODAG

---

**Input:** MEC $\mathcal{E}$

1   $\mathcal{E}' \leftarrow \mathcal{E}$;
2   **while** *there exists an undirected edge in* $\mathcal{E}$ **do**
3     Choose a vertex $V$ in $\mathcal{E}'$ such that (i) $V$ is a sink node in $\mathcal{E}'$ (no outgoing directed edges), and
      (ii) all undirected neighbors of $V$ are pairwise adjacent in $\mathcal{E}'$ (form a clique);
4   **foreach** *undirected edge* $(U, V)$ *in* $\mathcal{E}$ **do**
5     Orient $(U, V)$ as $U \to V$ in $\mathcal{E}$;
6   Remove $V$ from $\mathcal{E}'$;
7   **return** $\mathcal{E}$

---

---

**Algorithm 8:** Less Greedy Equivalence Search-0 (LGES-0)

---

**Input:** Data $\mathbf{D} \sim \mathbf{P}(\mathbf{v})$, scoring criterion $S$, prior assumptions $\mathbf{S} = \langle \mathbf{R}, \mathbf{F} \rangle$, initial MEC $\mathcal{E}_0$,
      insertion strategy $GetInsert$ in $\{$GETSAFEINSERT, GETCONSERVATIVEINSERT$\}$
**Output:** MEC $\mathcal{E}$ of $\mathbf{P}(\mathbf{v})$

1   $\mathcal{E} \leftarrow \mathcal{E}_0$ ;                      `// allows initialisation if preferred by user`
2   **repeat**
3     $\mathcal{G} \leftarrow$ some DAG in $\mathcal{E}$;                           `// forward phase`
4     $priorityList \leftarrow$ GETPRIORITYINSERTS$(\mathcal{E}, \mathcal{G}, \mathbf{S})$;
5     **foreach** *candidates in* $priorityList$ **do**
6       $(X_{max}, Y_{max}, \mathbf{T}_{max}) \leftarrow GetInsert(\mathcal{E}, \mathcal{G}, \mathbf{D}, candidates, S)$;
7       **if** $(X_{max}, Y_{max}, \mathbf{T}_{max})$ *is found* **then**
8         $\mathcal{E} \leftarrow \mathcal{E} +$ INSERT$(X_{max}, Y_{max}, \mathbf{T}_{max})$;
9         break ;                     `// no need to check lower priority`
10   **until** *no improving insertions exist*;
11   **repeat**
12     $\mathcal{E} \leftarrow \mathcal{E} +$ highest-scoring TURN$(X, Y, \mathbf{T})$ ;            `// turning phase`
13   **until** *no score-increasing reversals exist*;
14   **repeat**
15     $\mathcal{E} \leftarrow \mathcal{E} +$ highest-scoring DELETE$(X, Y, \mathbf{T})$ ;         `// backward phase`
16   **until** *no score-increasing deletions exist*;
17   **return** $\mathcal{E}$

---

**Algorithm 9:** Less Greedy Equivalence Search+ (LGES+)

---

**Input:** Data $\mathbf{D} \sim \mathbf{P}(\mathbf{v})$, scoring criterion $S$, prior assumptions $\mathbf{S} = \langle \mathbf{R}, \mathbf{F} \rangle$, initial MEC $\mathcal{E}_0$,
      insertion strategy $GetInsert$ in {GETSAFEINSERT, GETCONSERVATIVEINSERT}
**Output:** MEC $\mathcal{E}$ of $\mathbf{P}(\mathbf{v})$

1   $\mathcal{E} \leftarrow \text{LGES}(\mathbf{D}, S, \mathbf{S}, \mathcal{E}_0, GetInsert)$ ;
2   $\mathcal{D} \leftarrow$ all deletions valid for $\mathcal{E}$;
3   **while** $|\mathcal{D}| > 0$ **do**
4      $(X_{max}, Y_{max}, \mathbf{T}_{max}) \leftarrow$ highest-scoring deletion in $\mathcal{D}$;
5      $\mathcal{E}' \leftarrow \mathcal{E} + \text{DELETE}(X_{max}, Y_{max}, \mathbf{T}_{max})$;
6      $\mathcal{E}' \leftarrow \text{LGES}^*(\mathbf{D}, S, \mathbf{S}, \mathcal{E}_0, GetInsert, X_{max}, Y_{max})$;
       // LGES* is a modified version of LGES that excludes edge insertions
         between the given $X, Y$
7      **if** $S(\mathcal{E}', \mathbf{D}) > S(\mathcal{E}, \mathbf{D})$ **then**
8        $\mathcal{E} \leftarrow \mathcal{E}'$;
9        $\mathcal{D} \leftarrow$ all deletions valid for $\mathcal{E}$;
10     **else**
11       $\mathcal{D} \leftarrow \mathcal{D} \setminus \{(X_{max}, Y_{max}, \mathbf{T}_{max})\}$;
12 **return** $\mathcal{E}$

---

# D   Experiments

**Compute details.** All experiments were run on a shared compute cluster with 2x Intel Xeon Platinum 8480+ CPUs (112 cores total, 224 threads) at up to 3.8 GHz, and 210 MiB L3 cache.

## D.1   Learning from observational data

**Baseline details** We ran the PC algorithm using significance level $\alpha = 0.05$ for conditional independence tests, with the null hypothesis of independence. Since NoTears often outputs cyclic graphs, we post-processed the output graph by greedily removing the lowest-weight edges until it was acyclic, following [37]. This was done so that we could convert the output to a valid CPDAG for comparison with other algorithms. We ran NoTears with default parameters from the `causalnex` library, which uses a weight threshold of $w = 0$; note that performance may vary depending on the choice of parameters, particularly the weight threshold and the acyclicity penalty parameter. However, since the implementation of NoTears is very compute-heavy, we were not able to tune these parameters. For fair comparison, we implemented all GES variants, including XGES, LGES, and GIES, by modifying the code provided by [20].[5] Discrepancies in runtime of XGES between our results and the results of [36] may result from the latter's use of an optimized C++ implementation, which includes an implementation of the search operators that they used in XGES but not in GES.

### D.1.1   Synthetic data details for Experiment 5.1

For the results shown in Fig. 4, we draw Erdős–Rényi graphs with $p$ variables and {$2p$ edges in expectation (ER2), for $p \in \{5, 10, 15, 20, 25, 35, 50, 75, 100, 150\}$. For each $p$, we sample 50 graphs and generate linear-Gaussian data for each graph. Following [37], we draw weights from $\mathcal{U}([-2, -0.5] \cup [0.5, 2])$. In some cases, the resulting weight matrix was (almost) singular, in which case each row of weights was $\ell_1$ normalized. We draw noise means from $\mathcal{N}(0, 1)$ and noise variances from $\mathcal{U}([0.1, 0.5])$. We obtain $n = 10^4$ samples per dataset via `sempler` [20].

### D.1.2   Further baselines and metrics for Experiment 5.1

In Fig. D.1.1, we present additional baselines and accuracy metrics for the setting in Sec. 5.1, including the particular types of structural errors (excess adjacencies, missing adjacencies, incorrect orientations). As in the case of SHD, LGES outperforms other algorithms across these metrics, with CONSERVATIVEINSERT outperforming SAFEINSERT.

---

[5] `https://github.com/juangamella/ges`

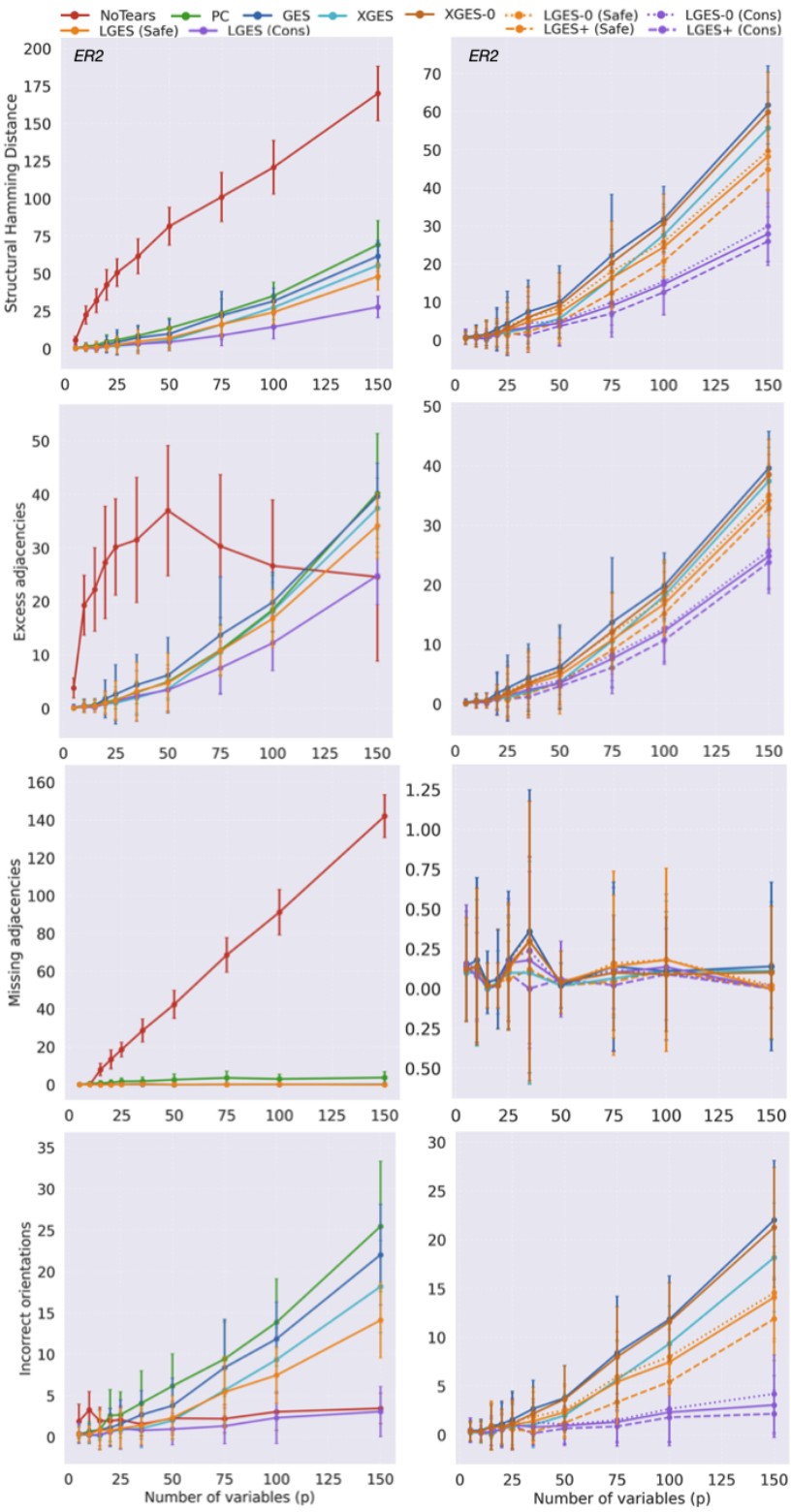

Figure D.1.1: Structural error of algorithms on $50$ simulated observational datasets from Erdős–Rényi graphs with $p$ variables and $2p$ edges in expectation, given $n = 10^4$ samples and no prior knowledge (Sec D.1.2). **Lower is better** (more accurate / faster) across all plots. **(Left)** Common baselines. GES-style algorithms outperform PC and NoTears in accuracy. **(Right)** GES variants. LGES-0, LGES, and LGES+ are the less greedy variants of GES, XGES-0, and XGES respectively. Each less greedy algorithm improves on its greedy counterpart in accuracy.

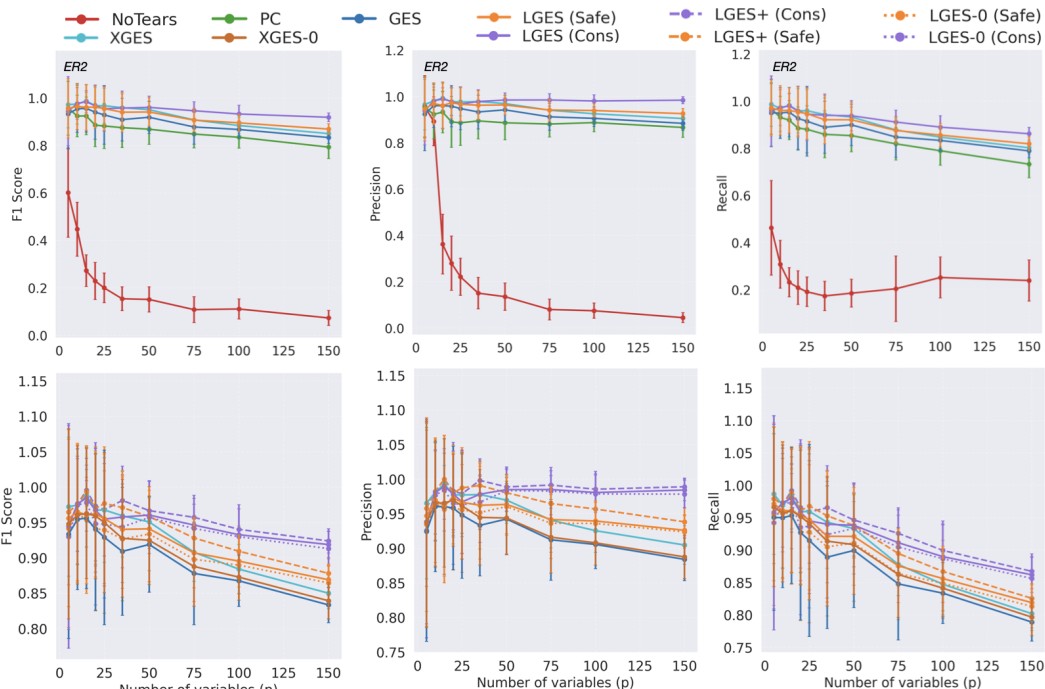

Figure D.1.2: Binary classification accuracy of algorithms on 50 simulated observational datasets from Erdős–Rényi graphs with $p$ variables and $2p$ edges in expectation, given $n = 10^4$ samples and no prior knowledge (Sec. D.1.3). **Higher is better** across all plots. An MEC is said to contain an edge $(X, Y)$ if it contains either $X - Y$ or $X \to Y$ (but not $Y \to X$). **(Top)** Common baselines. GES-style algorithms outperform PC and NoTears in accuracy. **(Bottom)** GES variants. LGES-0, LGES, and LGES+ are the less greedy variants of GES, XGES-0, and XGES respectively. Each less greedy variant improves on its greedy counterpart in accuracy.

The behaviour of NoTears is more variable. It misses significantly more adjacencies than all other methods—approximately linearly many in the number of variables. It also includes many more excess adjacencies than the other algorithms up to $p = 50$, after which the number of excess adjacencies begins to decline, ultimately approaching that of LGES for $p = 150$; however, this could be explained by the increasing number of adjacencies that are also missed by NoTears. We show in the next section how, as a result, the F1 score of NoTears is low across all graph sizes (Fig. D.1.2).

### D.1.3   Precision, recall, and F1 score for Experiment 5.1

Next, in addition to structural error, we evaluate accuracy by considering causal discovery as a binary classification task. We use the task definition provided in [36]: an MEC $\mathcal{E}$ is said to contain an edge $(X, Y)$ if it contains either $X \to Y$ or $X - Y$ (but not $Y \to X$). The results are shown in Fig. D.1.2. For graphs with $p < 20$ nodes, we find that GES and LGES have similar F1 scores. They both outperform PC, which in turn substantially outperforms NoTears. For $p > 20$, LGES+ (Conservative) dominates, followed by LGES (Conservative). For large $p$, LGES (Conservative) achieves a substantially higher F1 score than GES and other algorithms except LGES+. For instance, for $p = 150$, LGES (Conservative) achieves an F1 score just under 0.95, whereas GES's F1 score is slightly under 0.85.

### D.1.4   Further experiments with varying edge densities

In Fig. D.1.3, we provide results for select baselines on Erdős–Rényi graphs with $p$ variables and $\{1, 3\} \cdot p$ edges in expectation following the set-up in Experiment 5.1. The relative performance of algorithms is similar to the ER2 case (Fig. D.1.1), though PC and GES achieve similar accuracy on ER3 graphs. LGES+ (Conservative) is the most accurate overall, achieving $2\times$ less structural error than GES on ER1 and ER3 graphs. For instance, graphs with 150 variables and 450 edges in expectation, LGES (Conservative) achieves SHD $\approx 40$, and LGES+ (Conservative) $\approx 30$. All less

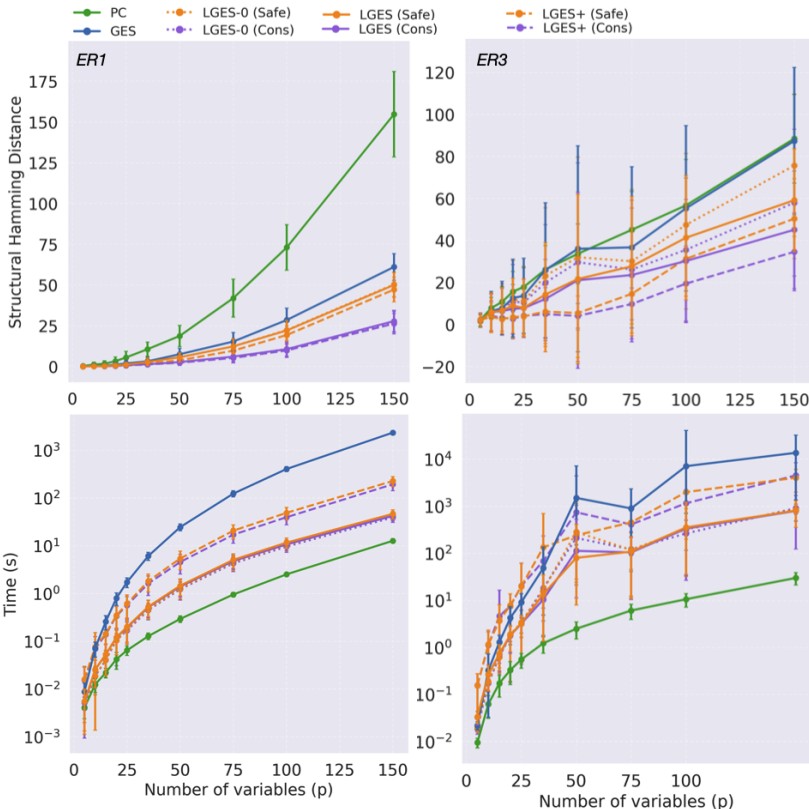

Figure D.1.3: Performance of algorithms on 50 simulated observational datasets from Erdős–Rényi graphs with $p$ variables and **(left)** $p$ edges and **(right)** $3p$ edges in expectation, given $n = 10^4$ samples and no prior knowledge (Sec. D.1.4). **Lower is better** (more accurate / faster) across all plots. See Fig. 4 for graphs with $2p$ edges in expectation. LGES (Conservative) is the fastest GES variant across edge densities, whereas LGES+ (Conservative) is the most accurate overall algorithm across edge densities.

greedy algorithms are faster and more accurate than GES, with LGES-0 and LGES being up to $10\times$ faster.

### D.1.5  Further experiments with smaller datasets

To investigate performance in the small-sample setting, we conduct experiments with $n = 10^3$ for ER2 graphs with $p \in \{10, 25, 50, 100\}$ variables. The results are summarized in Fig. D.1.4.

As in previous experiments, LGES (Safe and Conservative) outperforms GES in runtime and accuracy across all graph sizes, with CONSERVATIVEINSERT yielding higher accuracy than SAFEINSERT. Interestingly, the PC algorithm outperforms LGES for the case of $p = 100$, suggesting PC's usefulness for settings where the number of samples is small compared with the number of variables.

### D.1.6  Further experiments with larger graphs

We scaled up Experiment 5.1 to graphs with $p \in \{175, 250\}$ variables. We ran LGES (Safe and Conservative) and PC. We were unable to continue running XGES, GES, and NoTears due to time and compute constraints; for instance, GES took over $10^4$ seconds without terminating for $p = 175$ for a single trial. The results are summarized in Table D.1.1.

LGES both with SAFEINSERT and with CONSERVATIVEINSERT is substantially more accurate (in terms of SHD and F1 score) than PC, with CONSERVATIVEINSERT achieving less than half the structural error of PC. While PC is faster than LGES, LGES is much faster than GES, taking only $\approx$5-6 minutes for $p = 175$ and $\approx$15-20 minutes for $p = 250$. Recall that for $p = 150$, GES was already approaching a runtime of $10^4$ seconds (>2 hours) (Fig. 4a).

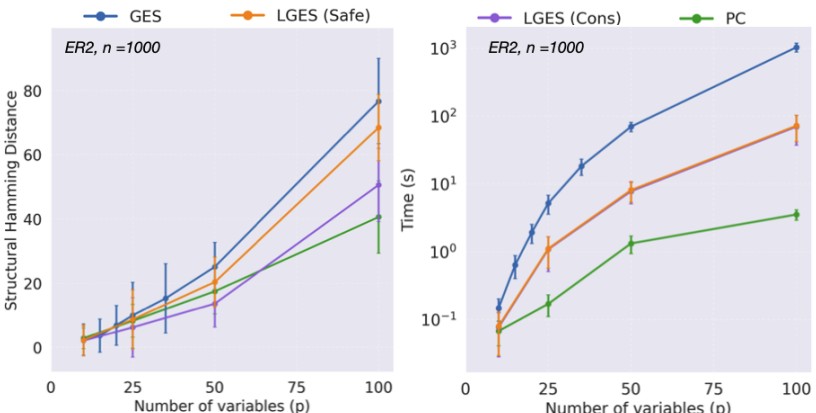

Figure D.1.4: Performance of algorithms on 50 simulated observational datstets from Erdős–Rényi graphs with $p$ variables and $2p$ edges in expectation, given the smaller dataset of $10^3$ samples and no prior knowledge (Sec D.1.5). In terms of accuracy, LGES (Conservative) outperforms other algorithms except for the case of $p = 100$, when PC is more accurate.

| Metric | Method | $p = 175$ | $p = 250$ |
|---|---|---|---|
| SHD | LGES-0 (Conservative) | $42.04 \pm 11.51$ | $83.06 \pm 14.42$ |
| | LGES (Conservative) | $\mathbf{39.457 \pm 10.44}$ | $\mathbf{82.80 \pm 14.73}$ |
| | LGES-0 (Safe) | $70.62 \pm 11.68$ | $135.24 \pm 14.31$ |
| | LGES (Safe) | $67.65 \pm 12.31$ | $133.22 \pm 14.07$ |
| | PC | $91.34 \pm 17.17$ | $166.20 \pm 24.34$ |
| Runtime | LGES-0 (Conservative) | $315.83 \pm 73.23$ | $989.37 \pm 205.53$ |
| | LGES (Conservative) | $326.44 \pm 73.166$ | $1019.13 \pm 198.12$ |
| | LGES-0 (Safe) | $337.45 \pm 78.43$ | $1058.11 \pm 221.32$ |
| | LGES (Safe) | $348.23 \pm 80.57$ | $1095.71 \pm 209.10$ |
| | PC | $\mathbf{29.20 \pm 9.50}$ | $\mathbf{85.67 \pm 23.52}$ |

Table D.1.1: Performance of algorithms (mean $\pm$ std) on 50 simulated observational datasets from large Erdős–Rényi graphs with $p$ variables and $2p$ edges in expectation, given $n = 10^4$ samples and no prior knowledge (Sec. D.1.6). **Lower is better** (more accurate / faster) across all metrics. Other algorithms are omitted as they exceeded the $10^4$ second bound on runtime for these graph sizes, prohibiting repeated trials. Among these, PC is the fastest, but makes twice as many structural errors as LGES (Conservative).

## D.2   Learning with prior knowledge

In Fig. D.2.1, we present more detailed plots of runtime and SHD for the setting in Sec. 5.2, including background knowledge comprising $m' = m/2$ and $m' = 3m/4$ edges for a ground truth graph with $m$ edges. Resutls for $m' = m/2$ follow a very similar trend to those with $m' = 3m/4$ discussed in Sec. 5.2. Initialisation outperforms prioritization in LGES when the expert knowledge is mostly correct ($\texttt{fc} \geq 0.75$), but prioritization does better otherwise.

## D.3   Learning from interventional data

In this section, we further discuss the performance of algorithms given interventional data in Sec. 5.3. We observed that LGES (Safe and Conservative) followed by $\mathcal{I}-$ORIENT has higher SHD from the ground truth than LGIES, Safe and Conservative respectively. We hypothesize that this is because, in our experiments, LGES only uses $10^4$ observational samples during the MEC learning phase. It uses the interventional samples only to orient edges using $\mathcal{I}$-ORIENT. In contrast, LGIES uses $10^4 + k \cdot 10^3$ samples in the MEC learning phase given $k$ interventions, since it uses interventional

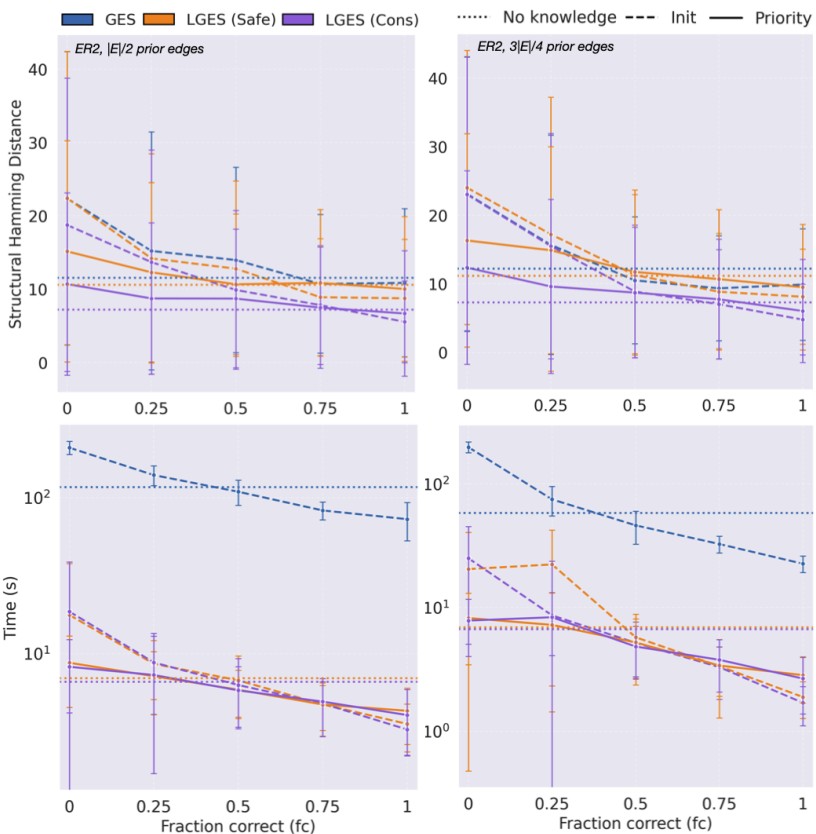

Figure D.2.1: Performance of algorithms on 50 simulated observational datasets from Erdős–Rényi graphs with 50 variables and 100 edges in expectation given $n = 10^3$ samples (Experiment 5.2, Sec. D.2). **Lower is better** (more accurate / faster) across all plots. We vary the correctness of the prior knowledge on the x-axis; higher `fc` indicates a higher fraction of knowledge that is correct. **(Left)** Algorithms are given $m' = m/2$ required edges as knowledge, for a true graph containing $m$ edges. **(Right)** Similar, but with $m' = 3m/4$. LGES' prioritization strategy is more robust to misspecification in the knowledge ($\leq 75\%$ correct) than initialization with the same knowledge

data throughtout learning, and we generate $10^3$ interventional samples per target. Moreover, since we choose $k = p/10$ (where $p$ is the number of variables), LGIES is making use of much more data than LGES for learning the MEC. Although GIES is known not to be asymptotically correct [58], this further motivates research into an asymptotically correct causal discovery algorithm that can make use of interventional data throughout learning.

### D.4 Real-world protein signaling data

**Sachs dataset.** We compare GES and LGES on a real-world protein signaling dataset [48]. The observational dataset consists of 853 measurements of 11 phospholipids and phosphoproteins. We compare the output of our methods with the gold-standard inferred graph [48, Fig. 3]), containing 11 variables and 17 edges. The dataset is continuous but violates the linear-Gaussian assumption.[6] We test our methods both on the original continuous dataset and on a discretized version (3 categories per variable corresponding to low, medium, and high concentration) from the `bnlearn` repository.[7]

**Results.** The learned graphs provided in Fig. D.4.1. All algorithms output the same MEC and thus have the same accuracy in both settings. With discrete data, each algorithm had an SHD of 9 edges from the reference MEC, all of which were missing adjacencies. With continuous data, each

---

[6] https://www.bnlearn.com/research/sachs05/
[7] https://www.bnlearn.com/book-crc-2ed/

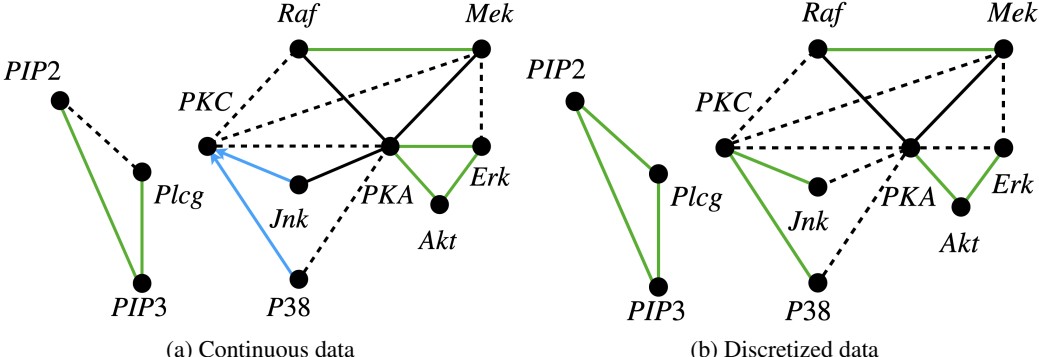

(a) Continuous data        (b) Discretized data

Figure D.4.1: Comparison between the reference MEC and the learned MEC for $n = 853$ observational samples from the Sachs protein-signaling dataset [48]. For both continuous and discretized data, the three algorithms (GES, LGES (SAFEINSERT) and LGES (CONSERVATIVEINSERT)) return the same MEC, so only one learned graph is shown per panel. Green solid lines indicate edges correctly recovered by the algorithms. Blue solid lines indicate edges misoriented by the algorithms. Black dashed lines indicate edges missed by the algorithms. **(a)** Continuous data: nine edges are missed and two are misoriented: $Jnk \rightarrow Pkc$ and $P38 \rightarrow PKC$, both undirected in the reference MEC. **(b)** Discretised data: nine edges are missed and none are misoriented.

algorithm had an SHD of 11 edges from the reference MEC, 9 of which were missing adjacencies and 2 of which were incorrect orientations.

# E    Frequently Asked Questions

Q1. What is the difference between score-based and constraint-based causal discovery?

**Answer.** Constraint-based and score-based approaches to causal discovery solve the same problem but in different ways. Constraint-based approaches such as PC [54] and Sparsest Permutations [44] use statistical tests, usually for conditional independence, to learn a Markov equivalence class from data. Score-based approaches such as Greedy Equivalence Search [9, 35] instead attempt to maximize a score (for e.g., the Bayesian Information Criterion or BIC [51]) that reflects the fit between graph and data. There is no general claim about which of these methods is superior; we refer readers to [56] for an extensive empirical analysis, who found, for instance, that GES outperforms PC in accuracy across various sample sizes [56, Tables 4, 5].

Q2. I thought causal discovery was only one problem, but it seems the paper claims to be solving three different tasks: observational observational learning, interventional learning, learning with prior knowledge. Can you elaborate on these tasks (why they are different, how they relate, etc.)?

**Answer.** The most well-studied problem in causal discovery is that of learning causal graphs from only observational data. However, algorithms for this task have a few limitations. Firstly, they are computationally expensive and often fail to produce quality estimates of the true Markov equivalence class from finite samples. This motivates using prior knowledge in the search to produce better quality estimates of the true Markov equivalence class and do so faster. Secondly, algorithms for learning from observational data only identify a Markov equivalence of graphs, since this is the most informative structure that can be learned from observational data. However, MECs can be quite large, and thus uninformative for downstream tasks such as causal inference. When interventional data is available, more edges in the true graph become identifiable. This motivates using interventional data to identify a smaller and more informative interventional Markov equivalence class—the task of interventional learning. When no prior assumptions or interventional data are available, these tasks collapse to observational learning.

Q3. If GES is asymptotically consistent, why bother to consider LGES?

**Answer.** While GES is guaranteed to recover the true Markov equivalence class given infinite samples, it faces two challenges. First, computational tractability: structure learning

is an NP-hard problem, and GES commonly struggles to scale in high-dimensional settings. Second, data is often limited in practice, which results in GES failing to recover the true MEC. LGES improves on GES in both of these aspects; it is up to 10 times faster and 2 times more accurate (Experiment 5.1). Moreover, while GES and LGES can both incorporate prior assumptions to guide the search, LGES is more robust to misspecification in the assumptions (Experiment 5.2).

Q4. How well does LGES scale?

**Answer.** LGES can scale to graphs with hundreds of variables and is up to 10 times faster than GES (Sec. D.1). Both variants of LGES (Safe and Conservative) terminate in less than 20 minutes hours on graphs with 250 variables (Sec. D.1.6) and have substantially better accuracy than other baselines (including PC and NoTears). D.1.6). LGES also outperforms these baselines in settings with dense graphs (Sec. D.1.4).

Q5. LGES may be asymptotically correct, but how well does it perform with finite samples?

**Answer.** We conduct an extensive empirical analysis of how LGES performs compared with baselines in finite-sample settings. Both variants of LGES (SAFEINSERT and CONSERVA-TIVEINSERT) have substantially better accuracy than GES, PC, and NoTears in experiments with $n = 10^4$ samples and up to 500 variables (Experiments 5.1, D.1). For instance, in graphs with 150 variables and 300 edges in expectation, LGES with CONSERVATIVEIN-SERT only makes $\approx$30 structural errors on average, which is twice as accurate as GES, which makes $\approx$60 structural errors. LGES also outperforms GES in smaller sample settings ($n \in \{500, 1000\}$) (Experiment D.1.5).

Q6. What is the difference between SAFEINSERT and CONSERVATIVEINSERT?

**Answer.** LGES can use either the SAFEINSERT or the CONSERVATIVEINSERT strategy to select INSERT operators in the forward phase; both result in improved accuracy and runtime relative to GES. We show that LGES with SAFEINSERT is asymptotically guaranteed to recover the true MEC (Cor. 1). However, it remains open whether the same is true of LGES with CONSERVATIVEINSERT, though we provide partial guarantees (Prop. C.1, C.2). Both strategies result in similar runtime, though CONSERVATIVEINSERT consistently has greater accuracy than SAFEINSERT across our experiments (Sec. 5, D).

Q7. How is causal discovery with prior knowledge different from initializing a causal discovery task to the hypothesized model?

**Answer.** Causal discovery with knowledge is the more general problem of using possibly misspecified prior knowledge in the process of causal discovery to aid the search. Initializing the search to a tentative model is one way to achieve this. However, initialization is not robust to misspecification in the assumptions and can result in worse runtime and accuracy than our approach of guiding the search using prior knowledge (Sec. 3.3, Experiment 5.2).

Q8. What is the difference between Greedy Interventional Equivalence Search (GIES) [24] and LGES + $\mathcal{I}$-ORIENT?

**Answer.** GIES and LGES are both score-based algorithms for learning from a combination of observational and interventional data. However, LGES has a few primary advantages over GIES. First, LGES with SAFEINSERT is guaranteed to recover the true interventional MEC (Cor. 1, Thm. 2) in the sample limit whereas GIES is not [58]. Second, both variants of LGES are up to 10 times faster than GIES, and LGES with CONSERVATIVEINSERT has accuracy competitive with GIES (Experiment 5.3). However, LGES uses interventional data to only to orient edges in a learned Markov equivalence class (in the $\mathcal{I}$-ORIENT procedure, Alg. 2), whereas GIES uses a combination of observational and experimental data throughout. We additionally introduce LGIES, which incoporates less greedy insertion into GIES and is thus both faster and more accurate than GIES (Experiment 5.3).

Q9. Can your work be combined with other causal discovery algorithms?

**Answer.** Several components of our approach are modular and can be combined with other causal discovery algorithms. For one, other algorithms in the GES family like FGES [43] and SGES [12] can be easily modified to use the SAFEINSERT or CONSERVATIVEINSERT strategies that we introduce; a simple extension would investigate the resulting changes in accuracy and runtime. For another, the $\mathcal{I}$-ORIENT procedure can be used to refine the observational MEC output by any causal discovery algorithm (not necessarily LGES) using interventional data.

