# OpenReview forum: "Less Greedy Equivalence Search"
_NeurIPS.cc/2025/Conference — NeurIPS 2025 poster_

### Official Review · Reviewer_PUwW · 2025-06-22

**Clarity:** 4
**Significance:** 3
**Originality:** 3
**Rating:** 5
**Confidence:** 3

**Summary:**

The paper presents a Generalized- (GGES) and two variants of Less Greedy Equivalence Search (LGES) algorithms that hold identifiability results in a generalized setting, by modify the edge insertions step of the well-known GES algorithm. In a first step the authors prove that under particular conditions choosing any score-increasing edge insertion within GES still leads to identifiability of the true Markov Equivalence Class (MEC), even if this does not correspond to the most score-increasing edge insertion. They, furthermore, reason that performing particular edge insertions between two variables, A-B, with the largest score increase might not be optimal in the presence of similar, score decreasing insertion operations on A-B under the same MEC. In consequence two approaches are presented: a 'conservative' variant that only selects edge insertions with no score-decreasing insert throughout the whole MEC, and a 'safe' variant the considers score-decreases from an arbitrary DAG sampled from the MEC.

The authors furthermore prove that GGES can start from arbitrary MEC, and also softly incorporate and correct for (possibly misspecified) prior knowledge, dividing given knowledge into four categories of (weakly) required and forbidden knowledge, that get prioritizes in the search, but ignored if inconsistent with the current MEC. Finally, they a variant for the incorporation of interventional data for LGES via a novel "I-Orient" algorithm is presented that identifies an the I-MEC, based on additional interventional data.

Experiments are performed on Erdős–Rényi graphs with p variables 2p edges and up to 150 variables on data of randomly sampled additive linear Gaussian noise models. The presented experimental results on the synthetic ER graphs indicate consistent superior performance over the compared to GES and GIES algorithms for observational and interventional data in terms of inference performance (SHD, Precision, Recall) and overall runtime. Similarly, the authors show in the appendix on-par performance for real-world protein signaling data between their LGES and the standard GES algorithm. Similarly, the recovery from misspecified knowledge shows good recovery rates over GIES.

**Questions:**

1) Could the authors provide further insights on the changes related to enabling repeated phase application with all LGES and GES-type baselines? In particular, can GES baselines overall improve and is the GES-Init algorithm more able to remove and recover from misspecified edges?
2) Similarly, could the authors provide further insights in comparison to the performance under the possibly more competitive xGES algorithm? Could the presented LGES itself benefit from interleaved phase applications?
3) In corollary 2, the authors utilize the property of SafeInsert to operate on arbitrary DAGs of the current MEC in order to consider possible score-decreases. I assume that the corresponding DAG is sampled randomly? Could the authors confirm this or explain their method for choosing a particular DAG?

**Ethical Concerns:**

["NO or VERY MINOR ethics concerns only"]

**Final Justification:**

The authors provided good experimental comparisons with the XGES-0 and XGES methods, which I consider a generally competitive GES variant. The new results position the proposed LGES method well within existing work and seem to show competitiveness of the method. The clarifications with regard to runtime measurements seem plausible.

**Limitations:**

yes

**Quality:**

3

**Strengths And Weaknesses:**

The paper provides overall significant theoretical results and insights which do result in seemingly large empirical performance increases, and might have lead to further developments on edge-selection strategies beyond this paper. While I do not believe, that conclusions drawn from experiments will largely change in qualitaty, my main critique is the omission of a possibly recent competitive xGES baseline and the decision to not consider the common choice of phase repetition within GES baselines, (but also the proposed LGES itself).

**Strengths**

1) The paper is generally well written. The authors provide clear and convincing arguments, towards weaknesses of the original GES algorithm, that limit its efficiency and performance. The modification towards the GGES and LGES variants, are well described. The corresponding proofs for the various theorems and propositions towards correctness and MEC identifiability of the algorithms are well described (as they partly align the original GES proofs) and, to the best of my understanding, seem to be correct.
2) The theoretical proofs for MEC identifiability under arbitrary score-increasing edge insertion operations provides a novel and significant generalization for the choice of edge selection strategies of GES-class algorithms which might initiate to further developments on this topic.
3) The experiments are well setup and follow standard procedure on Erdős–Rényi (ER) graphs with sufficient variation on the number of variables, applied metrics and repetitions. The authors are able to demonstrate superior performance of both LGES variants in comparison to the chosen baselines (PC,GES,GIES,NOTEARS) over all experiments in the main paper for MEC identifiability for observational and interventional data, as well as the recovery from misspecified knowledge, in terms of performance and runtime. An additional evaluation on real-world protein signaling data indicates the same performance as the GES algorithm.



**Weaknesses**

1) While the authors discuss and cite recent this recent GES variant, they do not compare to Extremely Greedy GES (xGES, [1]), which performs insert/delete/turning operations in an interleaved sequence and has also shown strong improvements over the original GES. While this approach is unlikely to improve in runtime, it might pose a more competitive baseline in terms of overall MEC inference.
2) While the authors make use of mention the additional use of turning phase of [2], the work additionally suggests the repeated application of the forward, backward and turning phases until no further improvements in score can be observed. The presented pseudo code suggests that such repeated applications are not performed for LGES, and (if I did not miss the particular spot) the corresponding parameter in the code is set to false for all GES-like algorithm calls. While the presented results on a single iteration performance show remarkable performance increases, comparisons to GES/GIES with repeated phases might provide for a more competitive comparison. This particularly applies for the recovery of misspecified edges from prior knowledge, where the GES-Init algorithm can have the ability to insert novel edges after the, otherwise final, edge removal phase which is the otherwise only chance to eliminate erroneous edges.



**Minor Points**

* I would like to recommend to briefly mention the meaning of $Nd^G_Y$ in the background section.
* Additionally, I believe the term $nd^G_Y$ in line 217 should be capitalized and made bold, similar to the other notation.



[1] Nazaret, Achille, and David Blei. "Extremely greedy equivalence search." *arXiv preprint arXiv:2502.19551* (2025).

[2] Hauser, Alain, and Peter Bühlmann.  "Characterization and greedy learning of interventional Markov  equivalence classes of directed acyclic graphs." *The Journal of Machine Learning Research* 13.1 (2012): 2409-2464.

---

> ### Author Rebuttal · Authors · 2025-07-30
>
> Thank you for your perceptive comments and suggestions.
>
> # Q1, W2
> *Can GES/LGES benefit from repeated phase applications?*
> Thank you for raising this valuable point. Indeed, our submitted experiments did not include repeated phase applications. In response to your suggestion, we ran GES and both variants of LGES with repeated applications of the forward, backward, and turning phases [4] on ER graphs of average degree 2 with to 150 variables. We found that **repeated phases modestly improved the accuracy of GES/LGES (both variants). However, even without repeats, both variants of LGES significantly outperformed GES with repeats**. Please see summary results below. We will note this in our paper.
>
> # Q2, W1
> *How do interleaved backward/turning/forward phases in XGES compare with and improve on LGES?*
> Thank you for this insightful question.
> 1. To faciliate a fair comparison and shared implementation across GES variants, we implemented the XGES-0 [5] heuristic of prioritizing deletions before reversals before insertions in our Python implementation of GES/LGES. We again evaluated these algorithms on ER2 graphs with up to 150 variables. The resulting ranking by accuracy was: GES < XGES-0 < LGES (Safe) < LGES (Safe) + XGES-0 < LGES (Conservative) < LGES (Conservative) + XGES-0. Runtime followed the reverse order, but with LGES (Safe) + XGES-0 and LGES (Conservative) swapped.
> 2. To elaborate, we found that the XGES-0 heuristic modestly improved the accuracy and runtime of each algorithm. This is consistent with [5], when comparing their GES-r (our GES) and XGES-0. **Even without the XGES-0 heuristic, both variants of LGES significantly outperform XGES-0.**
> 3. XGES extends XGES-0 by forcing edge deletions and restarts. This causes a ~10x slowdown relative to XGES-0 (Fig. 4 [5]), making it less practical for integration with LGES in high-dimensional settings.
>
> **SHD**
> | Algorithm                     | 25            | 50            | 100            | 150             |
> |------------------------------|---------------|---------------|----------------|-----------------|
> | GES                          | 4.67 ± 1.89    | 58.22 ± 15.05  | 841.76 ± 102.80 | 4509.60 ± 1047.57 |
> | GES (Repeats)                | 4.11 ± 1.43    | 52.56 ± 13.64  | 767.14 ± 94.54  | 4060.84 ± 380.35  |
> | XGES-0                       | 3.85 ± 0.96    | 50.69 ± 10.94  | 759.37 ± 92.51  | 4043.42 ± 332.04  |
> | LGES (Safe)                  | 1.28 ± 0.84    | 8.81 ± 3.91    | 69.71 ± 22.33   | 251.39 ± 65.19    |
> | LGES (Safe, Repeats)         | 1.00 ± 0.66    | 6.73 ± 2.98    | 52.08 ± 16.48   | 192.57 ± 47.90    |
> | XGES-0 + LGES (Safe)         | 1.07 ± 0.51    | 7.40 ± 3.16    | 55.44 ± 17.30   | 208.16 ± 51.64    |
> | LGES (Conservative)          | 1.17 ± 0.59    | 8.39 ± 3.68    | 66.92 ± 21.82   | 240.40 ± 62.26    |
> | XGES-0 + LGES (Conservative) | 1.06 ± 0.57    | 7.06 ± 2.99    | 53.33 ± 17.18   | 198.33 ± 49.03    |
> | LGES (Conservative, Repeats) | **0.95 ± 0.50**    | **6.56 ± 2.86**    | **50.65 ± 16.31**   | **185.35 ± 45.55**    |
>
> **Time**
>
> | Algorithm                     | 25            | 50            | 100            | 150             |
> |------------------------------|---------------|---------------|----------------|-----------------|
> | GES                          | 4.67 ± 1.89    | 58.22 ± 15.05  | 841.76 ± 102.80 | 4509.60 ± 1047.57 |
> | GES (Repeats)                | 4.11 ± 1.43    | 52.56 ± 13.64  | 767.14 ± 94.54  | 4060.84 ± 380.35  |
> | LGES (Safe)                  | 1.28 ± 0.84    | 8.81 ± 3.91    | 69.71 ± 22.33   | 251.39 ± 65.19    |
> | LGES (Safe, Repeats)         | 1.00 ± 0.66    | 6.73 ± 2.98    | 52.08 ± 16.48   | 192.57 ± 47.90    |
> | LGES (Conservative)          | 1.17 ± 0.59    | 8.39 ± 3.68    | 66.92 ± 21.82   | 240.40 ± 62.26    |
> | XGES-0                       | 3.85 ± 0.96    | 50.69 ± 10.94  | 759.37 ± 92.51  | 4043.42 ± 332.04  |
> | XGES-0 + LGES (Safe)         | 1.07 ± 0.51    | 7.40 ± 3.16    | 55.44 ± 17.30   | 208.16 ± 51.64    |
> | XGES-0 + LGES (Conservative) | 1.06 ± 0.57    | 7.06 ± 2.99    | 53.33 ± 17.18   | 198.33 ± 49.03    |
> | LGES (Conservative, Repeats) | **0.95 ± 0.50**    | **6.56 ± 2.86**    | **50.65 ± 16.31**   | **185.35 ± 45.55**  |
>
>
> # Q3
> *How does SafeInsert choose a DAG in the current MEC?*
> 1. Thank you for your careful reading. We do not sample the DAG randomly, but use a simple polynomial-time algorithm for converting a PDAG to DAG presented in [1] and used in the original GES [2]. It works as follows.
>     - Given a PDAG $\mathcal{G}$, copy it to $\mathcal{G}'$.
>     - Choose some vertex $V$ in $\mathcal{G}'$ such that
>         1. $V$ is a sink node (no outgoing directed edges) in $\mathcal{G}'$
>         2. all undirected neighbors of $V$ are adjacent to each other in $\mathcal{G}'$.
>     - Direct all edges in $\mathcal{G}$ into $V$.
>     - Remove $V$ from $\mathcal{G}'$.
>     - Repeat with another $V$ and $\mathcal{G}'$, until all edges in $\mathcal{G}$ are directed.
>
> Condition 1 ensures that orienting edges into $V$ will not create any cycles. Condition 2 ensures that orienting edges into $V$ will not create any new v-structures (aka unshielded colliders).
>
> 2. In theory, any procedure for choosing a DAG from an MEC suffices. There is a poly-time algorithm for uniform random sampling of a DAG from an MEC [3], with C++/Julia implementations available. We are open to investigating how the choice of DAG affects SafeInsert performance if the reviewers think this is a fruitful direction. Presumably, any SafeInsert variant would not be as performant as ConservativeInsert, which checks for the existence of any DAG witnessing a separation.
>
>
> [1] Dorit Dor and Michael Tarsi. A simple algorithm to construct a consistent extension of a partially oriented graph. Technical Report R-185, Cognitive Systems Laboratory, UCLA Computer Science Department, October 1992.
>
> [2] David Maxwell Chickering. Optimal structure identification with greedy search. J. Mach. Learn. Res., 3(null):507–554, March 2003.
>
> [3] Marcel Wienobst, Max Bannach, and Maciej Liskiewicz. Polynomial-Time Algorithms for Counting and Sampling Markov Equivalent DAGs with Applications. J. Mach. Learn. Res. 24(213):1-45, Jul 2023.
>
> [4] Alain Hauser and Peter Bühlmann. Characterization and greedy learning of interventional
> markov equivalence classes of directed acyclic graphs. J. Mach. Learn. Res., 13(1):2409–2464,
> Aug 2012.
>
> [5] Achille Nazaret and David Blei. Extremely greedy equivalence search. In Negar Kiyavash and Joris M. Mooij, editors, Proceedings of the Fortieth Conference on Uncertainty in Artificial Intelligence, volume 244 of Proceedings of Machine Learning Research, pages 2716–2745. PMLR, 15–19 Jul 2024.

---

> > ### Comment · Reviewer_PUwW · 2025-08-03
> >
> > Thank you for providing further experimental results, comparing LGES methods with XGES-0 and even integrating the GES-0 methodology into LGES. The additional clarification on the SafeInsert method, as well as the provided results tables[1], definitely raises the quality of the paper.
> >
> > While I find the additional evaluations insightful, I'm slightly confused about why the authors do not compare to the 'full' XGES approach. In the original paper XGES, in comparison to XGES-0, featured a quite sharp drop in terms of SHD even for small and moderately sized graphs (Fig. 2). I understand that runtime scalings of the algorithms might differ for different parameterizations of the benchmarked graphs, making them possibly unsuited for integration in LGES in terms of runtime. However, when compared to Fig. 4 of the XGES paper, run time scaling for both XGES and XGES-0 seemed to be similar or rather below that of GES, which the authors naturally do compare to in their evaluations. I still believe that including full XGES would give valuable insights in terms of absolute performance for applications on small- to medium-sized graphs or applications that aren't time-sensitive. While I'm still positive about recommending accepting the paper, I would still like to ask the authors to comment on this point.
> >
> > [1] Compared to the other replies, I believe the authors accidentally copied the timetable twice.

---

> ### Author Response · Authors · 2025-08-04
>
> Thank you for the positive remarks and for engaging with our new results.
>
> Regarding your note:
> >*“...why the authors do not compare to the 'full' XGES.”*
>
> We appreciate your suggestion. The initial omission was due to the effort required to implement full XGES fairly in our codebase, which was challenging within the rebuttal window. However, since submission, we have implemented full XGES and evaluated it on ER2 graphs of 25 and 50 nodes. Larger graphs are in progress and will be included in the revised manuscript.
>
> **In terms of accuracy, both insertion heuristics improve XGES.** While XGES outperforms XGES-0, the comparison with LGES (Conservative) is indecisive: LGES outperforms XGES for 50 nodes, while the opposite holds for 25 nodes, both by slim margins. XGES + LGES (Conservative) is the most accurate overall.
>
> **In terms of runtime, both heuristics provide up to a 10x speedup of XGES.** Without them, XGES is $\approx$30x slower than LGES and $\approx$4x slower than GES/XGES-0. Combining LGES with XGES slows LGES by 3x. XGES-0 + LGES (Conservative) is the fastest overall.
>
> **(Table 1) Runtime**
>
> |Algorithm|25|50|
> |---------|--|--|
> |XGES|17.25 ± 7.36|252.76 ± 131.06|
> |GES|4.67 ± 1.89|58.22 ± 15.05|
> |XGES-0|3.85 ± 0.96|50.69 ± 10.94|
> |XGES+LGES (Safe)|3.90 ± 2.68|27.74 ± 18.44|
> |XGES+LGES (Conservative)|3.84 ± 2.68|24.94 ± 11.96|
> |LGES (Safe)|1.28 ± 0.84|8.81 ± 3.91|
> |LGES (Conservative)|1.17 ± 0.59|8.39 ± 3.68|
> |XGES-0+LGES (Safe)|1.07 ± 0.51|7.40 ± 3.16|
> |XGES-0+LGES (Conservative)|1.06 ± 0.57|7.06 ± 2.99|
>
> We hope Table 2 below also addresses your note:
> >  *“...the authors accidentally copied the timetable twice.”*
>
> Thank you for catching this ($\dagger$).
>
> **(Table 2) SHD**
> |Algorithm|25|50|100|150|
> |---------|--|--|---|---|
> |GES|4.38 ± 8.45|10.00 ± 9.54|31.78 ± 8.59|61.78 ± 10.26|
> |GES (Repeats)|4.42 ± 8.43|10.14 ± 9.41|31.98 ± 8.51|62.39 ± 10.77|
> |XGES-0|3.04 ± 6.85|9.14 ± 8.38|30.66 ± 7.65|59.74 ± 10.10|
> |LGES (Safe)|3.62 ± 7.59|8.14 ± 9.48|25.80 ± 9.06|49.69 ± 10.02|
> |LGES (Safe, Repeats)|3.68 ± 7.61|8.20 ± 9.21|25.93 ± 8.97|50.37 ± 10.30|
> |XGES-0+LGES (Safe)|2.58 ± 5.80|7.14 ± 7.92|24.41 ± 7.97|47.42 ± 9.19|
> |XGES|2.02 ± 5.21|6.14 ± 4.09|||
> |LGES (Conservative)|3.06 ± 6.91|4.94 ± 6.58|15.39 ± 8.82|29.94 ± 9.45|
> |LGES (Conservative, Repeats)|3.10 ± 6.75|4.98 ± 6.66|15.70 ± 8.89|30.76 ± 9.65|
> |XGES-0+LGES (Conservative)|2.56 ± 5.95|4.42 ± 5.70|14.66 ± 7.91|26.95 ± 6.88|
> |XGES+LGES (Safe)|1.82 ± 3.65|4.98 ± 4.48|||
> |XGES+LGES (Conservative)|1.82 ± 3.84|3.74 ± 5.07|||
>
> Finally,  regarding your note that
> > *“Fig. 4 of the XGES paper, run time scaling for both XGES and XGES-0 seemed to be similar or rather below that of GES.”*
>
> We want to address why our runtime results seem to be in disagreement with those of [5].
>
> Like [5], we find that XGES-0 is faster than GES, but the speed-up is modest (see table in original rebuttal), unlike the $\approx$30x speedup reported in Fig. 4 [5]. Moreover, [5] reports that XGES is faster than GES, while we find that XGES to be $\approx$4x slower  (Table 1 above).
>
> To explain why, it’s important to emphasize that the **implementations of GES and XGES in [5] differ significantly from each other, and from ours.** [5] use a C++ implementation of XGES(-0) with optimized MEC operators. Our implementation, like theirs, changes the search heuristic, but does not include those optimizations. Importantly, the same optimizations could also accelerate GES, but were not used in [5], as far as we can tell.
>
> We elaborate on why XGES is naturally expected to be slower than GES, all else being equal. First, even in [5], XGES is $\approx$10x slower than XGES-0 (Fig. 4). As [5] notes, *“XGES repeatedly applies XGES-0,... one order of magnitude more score evaluations than XGES-0.”* This overhead is expected: XGES forces a deletion on the output of XGES-0, restarts XGES-0 from the resulting graph, and repeats. In the worst case, it may apply all possible deletes (up to exponentially many in the node degree) to a given MEC,  meaning exponentially many re-runs of XGES-0. Second, as stated, we find XGES-0 is only modestly faster than GES.
>
> Because our simulation setup closely mirrors that of [5], we believe the discrepancy in runtime arises primarily from implementation choices. For a fair comparison, full XGES would need to be reimplemented in our codebase (as we did) or vice versa.
>
> Our goal in this work is to advance the Pareto frontier in the accuracy/scalability tradeoff via new, principled search strategies. We believe we achieve this by providing two novel heuristics that improve both accuracy and scalability of existing GES variants. We hope this clarifies our decisions and are happy to incorporate more optimized implementations in a future version of the work.
>
>  ($\dagger$) Based on this table, we revise our earlier claim about repeated phase applications: they have no significant impact on the accuracy of GES/LGES.

---

> > ### Comment · Reviewer_PUwW · 2025-08-04
> >
> > Thank you for the repeated extensive effort in providing further experimental results and comparisons with XGES. The new results position the proposed LGES method well within existing work and seem to show competitiveness of the method. Clarifications with regard to runtime measurements seem plausible. I will raise my score to recommend a clear accept.

---

> > > ### Author Response · Authors · 2025-08-04
> > >
> > > Thank you for your generosity with your time in this constructive reviewing process. We appreciate your keen efforts to help us in improving the quality of our work and exposition.

---

### Official Review · Reviewer_AAo7 · 2025-06-30

**Clarity:** 3
**Significance:** 3
**Originality:** 3
**Rating:** 4
**Confidence:** 3

**Summary:**

This paper introduces Less Greedy Equivalence Search (LGES), which is an improved algorithm for causal discovery that improves upon the classic Greedy Equivalence Search (GES) method. It does so by doing doing edge selection less greedily, instead of greedily maximizing the score. LGES is shown to also leverage possibly misspecified prior assumptions and correct misspecified edges.

**Questions:**

I felt the paper was fairly clear and I liked how simulation was used to gain intuition on where GES may not do well. My only question is regarding a more extensive set of experiments, which in it of itself would reveal the algorithm's benefits. Finally, I will add that I'm also not super familiar with score based methods for CD and the associated literature, so perhaps other reviewers can champion the paper if they feel strongly about it.

**Ethical Concerns:**

["NO or VERY MINOR ethics concerns only"]

**Final Justification:**

My overall rating for the paper remains positive.

**Limitations:**

Please see weakness.

**Quality:**

3

**Strengths And Weaknesses:**

Strength:  The paper carefully develops two new strategies for improving on GES. As the authors wrote, it is the first asymptotically correct score-based procedure for learning from interventional data that can scale to graphs with more than a hundred nodes. LGES is also robust in handling misspecified graphs. And the authors do compare with other algorithms such as PC and NOTEARS, and not jus showing that LGES is only an improvement over GES.

Weaknesses: As one comment that stood out to me, I think the paper would be more compelling if a more extensive set of experiments was run. It would be more convincing if LGES was benchmarked on a real world dataset (perhaps Sachs which was considered in NOTEARS), as well as synthetic graphs of varying sparsity.

---

> ### Author Rebuttal · Authors · 2025-07-30
>
> We appreciate your time and thoughtful feedback on our work.
>
> # Q1
>  Please see below some new results regarding your comment,
>
> > I think the paper would be more compelling ... more extensive set of experiments… real world dataset (perhaps Sachs which was considered in NOTEARS), as well as synthetic graphs of varying sparsity.
>
> 1. **Real world data.** We indeed compared LGES (both variants) against GES using the Sachs dataset, as detailed in Appendix D.4. All algorithms achieve the same structural error, though LGES is faster. We would appreciate suggestions for additional real-world datasets to explore and feedback on the organization, if reviewers feel that this experiment should be highlighted in the main text.
> 2. **Synthetic graphs of varying sparsity.** Thank you for the helpful suggestion. We conducted additional experiments on Erdos Renyi graphs of average degree 1 and 3 (ER1, ER3 respectively) with up to 150 variables. Results are consistent with the ER2 case: LGES outperforms GES in both speed and accuracy. Summary results are shown below; we will add the relevant plots to the revised manuscript.
>
> **ER1 graphs, SHD**
>
> | Algorithm           | 25         | 50         | 100         | 150         |
> |--------------------|------------|------------|-------------|-------------|
> | GES                | 1.88 ± 1.98 | 7.52 ± 3.70 | 28.54 ± 7.26 | 61.10 ± 8.19 |
> | LGES (Safe)        | 1.16 ± 1.35 | 5.62 ± 2.84 | 22.40 ± 6.27 | 50.40 ± 7.58 |
> | LGES (Conservative)| **0.82 ± 1.65** | **2.68 ± 1.75** | **10.74 ± 4.37** | **27.52 ± 6.44** |
>
> **ER1 graphs, time**
> | Algorithm           | 25         | 50         | 100           | 150            |
> |--------------------|------------|------------|---------------|----------------|
> | GES                | 1.75 ± 0.41 | 24.70 ± 4.13 | 407.25 ± 48.87 | 2359.30 ± 209.12 |
> | LGES (Safe)        | 0.18 ± 0.10 | 1.29 ± 0.51 | 10.40 ± 2.54  | 43.29 ± 9.45   |
> | LGES (Conservative)| **0.17 ± 0.08** | **1.24 ± 0.51** | **9.67 ± 2.38**   | **39.54 ± 8.75**   |
>
>
> **ER3 graphs, SHD**
> | Algorithm           | 25          | 50          | 100          | 150          |
> |--------------------|-------------|-------------|--------------|--------------|
> | GES                | 13.66 ± 17.84 | 36.06 ± 48.99 | 55.34 ± 39.32 | 87.39 ± 34.99 |
> | LGES (Safe)        | 11.12 ± 16.57 | 32.00 ± 47.71 | 47.47 ± 33.98 | 75.67 ± 33.73 |
> | LGES (Conservative)| **10.44 ± 16.65** | **29.66 ± 47.32** | **35.57 ± 34.16** | **58.00 ± 34.97** |
>
> **ER3 graphs, time**
> | Algorithm           | 25          | 50            | 100            | 150             |
> |--------------------|--------------|---------------|----------------|-----------------|
> | GES                | 9.17 ± 4.88   | 1515.92 ± 5715.32 | 7166.08 ± 33723.75 | 13768.44 ± 18749.97 |
> | LGES (Safe)        | 3.40 ± 2.93   | 284.79 ± 996.03   | 326.55 ± 574.96    | **846.54 ± 487.28**     |
> | LGES (Conservative)| **3.29 ± 2.85**   | **212.79 ± 841.08**   | **266.61 ± 241.66**   | 914.51 ± 789.88 |
>
> 3. **Additional experiments varying the number of prior edge assumptions.** We varied the number of edges included in the background knowledge (Experiment 5.2). Again, results are similar: LGES outperforms GES-Init.
> 4. **Additional baselines.** To evaluate LGES more comprehensively, we conducted experiments with two new GES variants, as suggested by other reviewers: GES with repeated applications of the forward, backward, and turning phases (proposed in [1]), and the XGES-0 [2] heuristic of prioritizing deletions before insertions. We also combined LGES with these strategies. We evaluated all algorithms on ER2 graphs with up to 150 nodes. Findings include:
>     1. Repeated phase application modestly improved the accuracy of GES/LGES (both variants). However, **both variants of LGES (even without repeats) significantly outperform GES with repeats**.
>     2. The XGES-0 heuristic modestly improved the runtime and accuracy of GES and LGES (both variants). However, **both variants of LGES (even without the XGES-0 heuristic) significantly outperform XGES-0**. The resulting ranking by accuracy is: GES < XGES-0 < LGES (Safe) < LGES (Safe) + XGES-0 < LGES (Conservative) < LGES (Conservative) + XGES-0. Runtime followed the reverse order, but with LGES (Safe) + XGES-0 and LGES (Conservative) swapped.
>
> **SHD**
>
> | Algorithm                     | 25            | 50            | 100           | 150           |
> |------------------------------|---------------|---------------|---------------|---------------|
> | GES                          | 4.38 ± 8.45    | 10.00 ± 9.54   | 31.78 ± 8.59   | 61.78 ± 10.26  |
> | XGES-0                       | 3.04 ± 6.85    | 9.14 ± 8.38    | 30.66 ± 7.65   | 59.74 ± 10.10  |
> | LGES (Safe)                  | 3.62 ± 7.59    | 8.14 ± 9.48    | 25.80 ± 9.06   | 49.69 ± 10.02  |
> | XGES-0 + LGES (Safe)         | 2.58 ± 5.80    | 7.14 ± 7.92    | 24.41 ± 7.97   | 47.42 ± 9.19   |
> | LGES (Conservative)          | 3.06 ± 6.91    | 4.94 ± 6.58    | 15.39 ± 8.82   | 29.94 ± 9.45   |
> | XGES-0 + LGES (Conservative) | **2.56 ± 5.95**    | **4.42 ± 5.70**   | **14.66 ± 7.91**   | **26.95 ± 6.88**   |
>
>
>
> **Time**
> | Algorithm                     | 25            | 50            | 100            | 150             |
> |------------------------------|---------------|---------------|----------------|-----------------|
> | GES                          | 4.67 ± 1.89    | 58.22 ± 15.05  | 841.76 ± 102.80 | 4509.60 ± 1047.57 |
> | XGES-0                       | 3.85 ± 0.96    | 50.69 ± 10.94  | 759.37 ± 92.51  | 4043.42 ± 332.04  |
> | LGES (Safe)                  | 1.28 ± 0.84    | 8.81 ± 3.91    | 69.71 ± 22.33   | 251.39 ± 65.19    |
> | LGES (Conservative)          | 1.17 ± 0.59    | 8.39 ± 3.68    | 66.92 ± 21.82   | 240.40 ± 62.26    |
> | XGES-0 + LGES (Safe)         | 1.07 ± 0.51    | 7.40 ± 3.16    | 55.44 ± 17.30   | 208.16 ± 51.64    |
> | XGES-0 + LGES (Conservative) | **1.06 ± 0.57**    | **7.06 ± 2.99**    | **53.33 ± 17.18**   | **198.33 ± 49.03**    |
>
>
> [1] Alain Hauser and Peter Bühlmann. Characterization and greedy learning of interventional
> Markov equivalence classes of directed acyclic graphs. J. Mach. Learn. Res., 13(1):2409–2464,
> Aug 2012.
>
> [2] Achille Nazaret and David Blei. Extremely greedy equivalence search. In Negar Kiyavash and Joris M. Mooij, editors, Proceedings of the Fortieth Conference on Uncertainty in Artificial Intelligence, volume 244 of Proceedings of Machine Learning Research, pages 2716–2745. PMLR, 15–19 Jul 2024.

---

> > ### Comment · Reviewer_AAo7 · 2025-08-02
> > **Reply**
> >
> > Thank you to the authors for conducting the additional experiments, I appreciate it. I think the results give more evidence that the method is performant.
> >
> > As I mentioned in my review, I'm not an expert on this topic so it would be best for an expert to champion for the acceptance of this paper. But overall, I think positively of the paper. Thanks again.

---

> > > ### Author Response · Authors · 2025-08-04
> > >
> > > Thank you for taking the time to consider our updated results and for your encouraging comments in this constructive reviewing process.

---

### Official Review · Reviewer_RnrE · 2025-07-02

**Clarity:** 3
**Significance:** 4
**Originality:** 3
**Rating:** 5
**Confidence:** 3

**Summary:**

The authors propose a variant of GES that changes the forward phase. Most importantly, they show that if an Insert operation between X and Y decreases the score, then (some) other Insert operations between those variables do not need to be considered. Further, the algorithm allows the incorporation of background knowledge (but will ignore this if it contradicts the data) and can utilise interventional data. Experiments show this new algorithm is both faster and more accurate than the original GES.

**Questions:**

Q1: I could not quite follow the discussion in section 3.2 about a potential issue with ConservativeInsert. If I understand correctly, the issue is that sometimes, ConservativeInsert may discard the only score-increasing Insert(s). However, from Proposition 1, it seems that such score-increasing Inserts are not desirable because they create d-connections where we want d-separations. Can you give an example to show why it is nonetheless preferable to use a strategy that does consider such Inserts?

### Minor suggestions that don't impact my rating:
- on page 2, the period after "starts the search with $\mathcal{E}$" should be a comma
- I believe the "If not" in Strategy 2 should be "If so"
- in Example 3, $X$ should be $X_1$

**Ethical Concerns:**

["NO or VERY MINOR ethics concerns only"]

**Final Justification:**

I think this is a good paper with a valuable contribution, that deserves a place in the conference.

**Limitations:**

yes

**Quality:**

4

**Strengths And Weaknesses:**

This paper makes an important contribution, which is accompanied by theoretical proofs of correctness and thorough empirical evaluation. The paper is relevant both for researchers developing structure learning algorithms and for researchers wanting to apply them. Furthermore, the paper is clearly written.

---

> ### Author Rebuttal · Authors · 2025-07-30
>
> We greatly appreciate your feedback and positive assessment of our work.
>
> # Q1
>
> *On section 3.2’s discussion about a potential issue with ConservativeInsert.*
>
> Thank you for the perceptive question. We will revise our explanation to make this point more precise.
> 1. *"ConservativeInsert may discard the only score-increasing Insert(s)?"* Not exactly: it is currently unknown whether there exist cases where ConservativeInsert would (incorrectly) discard the only score-increasing Insert(s). If we can construct an example where it does, it would amount to proving that LGES + ConservativeInsert is theoretically unsound. If we can show no such example exists, it would establish soundness. Our work leaves the soundness of ConservativeInsert as an open question; despite our efforts, we were not able to prove or disprove it. Nonetheless, we include ConservativeInsert as a valuable heuristic, given its strong empirical performance relative to GES.
>
> 2. *"...such score-increasing Inserts are not desirable ...Can you give an example to show why it is nonetheless preferable to use a strategy that does consider such Inserts?"*
> It is difficult to construct a concrete example, as doing so would demonstrate that ConservativeInsert is not sound. However, our partial guarantees in Propositions C.1 and C.2 may serve as a starting point for proving soundness or giving an example that shows otherwise. To elaborate, these results imply that for any MEC $\mathcal{E}$ for which ConservativeInsert would discard the only score-increasing Insert(s), $\mathcal{E}$ would (a) already contain the skeleton of the true MEC, and (b) not disagree with the true MEC on the orientation of any unshielded triples they have in common. This may provide a useful direction for constructing a counterexample to soundness. Our empirical results suggest that such cases, if they exist, are at least rare in the ER model, since ConservativeInsert significantly outperforms both GES and SafeInsert.

---

> > ### Comment · Reviewer_RnrE · 2025-08-06
> >
> > Thanks for clarifying this point. I am also happy to see that, in response to the other reviews, other GES variants have been included in the related work and experiments, which further strengthens the paper. I maintain my positive rating.

---

> > > ### Author Response · Authors · 2025-08-06
> > >
> > > We appreciate your engagement with our clarifications and additional results. Thank you again for your comments -- they certainly help improve our work.

---

### Official Review · Reviewer_Bizy · 2025-07-02

**Clarity:** 3
**Significance:** 3
**Originality:** 2
**Rating:** 4
**Confidence:** 4

**Summary:**

It is a known practical issue that the GES algorithm may often produce denser graphs than truth with finite samples. This is partly because when an edge is added in the forward phase, only a single check of score increasing is performed. When this added edge is incorrect, the chance for it to be deleted in the backward phase is often small.

To deal with this issue, this paper proposes the algorithm "Less Greedy Equivalence Search" (LGES). Instead of adding an edge whenever a score increasing is obtained, LGES avoids adding edges even with increased scores whenever it implies **some (other)** conditional independences. Two variants of the insertion steps, named "Conservative Inserts" and "Safe Inserts" are proposed, one more aggressive with partial guarantees, and another with full guarantee.

Further, extensions to incorporate prior knowledge (even when some priors can be misspecified) and interventional data are provided.

**Questions:**

/

**Ethical Concerns:**

["NO or VERY MINOR ethics concerns only"]

**Final Justification:**

My concerns have been well addressed by the authors' response.

**Limitations:**

/

**Paper Formatting Concerns:**

/

**Quality:**

3

**Strengths And Weaknesses:**

**Strengths:**

1. The problem to tackle is quite relevant -- the practical issue of often-denser results given by GES indeed needs to be addressed.

2. The key motivation is clear and natural: in each step of the progression, it doesn't have to be the best score to be moved to. As long as the score is increasing it is a valid move. This principle leaves with space to strategically choosing among all candidates to get a balance between greediness and robustness.

3. The running examples used throughout is clear and helpful.


**Weaknesses:**

1. The implication of score decreasing to incorrect adjacency (Proposition 1) is sound and useful. **But is it complete (seems no)?** In other words, does the score decreasing in Proposition 1 capture **all** (or just some) possible conditional independence patterns that leads to non-adjacencies? Also, do the two insertion variants capture all what can be captured by Proposition 1? If the answers are no, the authors may need to justify why such heuristics is good enough in avoiding false adjacencies.

2. **The claim for speed-up needs more justification.** Intuitively, such more robust decision for adding edges comes with more tests: instead of just testing between X and Y given Y's current parents, now all possible neighbor sets T need to be tested (e.g., in conservative strategy). Then, why is LGES finally faster? Is it because that less edge is inserted and the graph in progression is sparser? Some kind of complexity analysis is expected.

3. **The backward phase (and the optional turning phase) is not optimized accordingly.** Intuitively the dual operations of choosing stable (not necessarily the best) neighbors can be applied to the backward phase to avoid false non-adjacencies. This also makes the whole algorithm more self contained and symmetric. I am curious why the authors didn't implement that. Is it because that it will usually hurt GES's performance?

4. **The contributions on dealing with interventional data and prior knowledge seem a bit irrelevant.**
   - For the interventional data, when one treats the experimental domain index as an additional root variable, the whole discovery method becomes nothing but to run plain GES (or any constraint based algorithms) on all variables including the domain index. Under this, I cannot see clearly why the authors need to separate this as a specific contribution. Or, are there any more specific optimization made to algorithmic procedure for interventional data?
   - For incorporating prior knowledge, the idea of seeing prior knowledge as priority constraints instead of hard initialization is interesting. (Though it is still worth discussing whether data or prior knowledge should be prioritized, or should be customizable to the users). Could the authors confirm whether this idea is newly proposed in this paper, or it is also already used in some existing works? Usually how do other score based variants incorporate background knowledge?
   - Overall, the technical discussions regarding these two aspects are too short in the main body. For example, the very important algorithm "GETPRIORITYINSERTS" is deferred to the appendix. And similarly many other references are deferred (e.g., Defs. A.3, A.4, A.5), making hyper links not work in main body. More detailed analysis of these two contributions are expected. They are now too vague.

5. **Many existing literature on GES are ignored.** There are many existing GES variants aiming to address different aspects of issues. For example,
   - "Statistically Efficient Greedy Equivalence Search" (http://proceedings.mlr.press/v124/chickering20a/chickering20a.pdf),
   - "Fast GES" (https://www.ccd.pitt.edu/wp-content/uploads/2018/10/FGES1c-user-documentation-5_21_2016-sample-size.pdf; this one, together with the point above that interventional GES is nothing different than plain GES, make the claim on "the first ... for learning from interventional data that can scale to ... more than a hundred node" suspicious),
   - "Extremely Greedy Equivalence Search" (https://proceedings.mlr.press/v244/nazaret24a.html; this is cited in the paper but not compared), etc.

   Could the authors please have a subsection reviewing those existing variants of GES, and in the experimental section, compare with them all? This is important since the purpose of this paper is for practical gain.

6. **The SafeInsertion strategy needs some more elaboration:**
   - "nd_Y^G" seems undefined?
   - Is it enough to pick up only one arbitrary DAG from the CPDAG, or should we iterate all DAGs (to find the highest score and to ensure the existence of a score increasing neighbor)?
   - "Check if G has a higher score .. if not, discard ..": then what if the score is unchanged, or an unchanged score is provably impossible (even with possibly misspecified graphs)?
   - At the first glance the SafeInsertion strategy looks very similar to the original strategy in GES: pick DAGs from CPDAG and then check if X is independent of Y conditioned on Y's current parents. Could the authors highlight the biggest difference there?



7. (Minor) Sentences like "This more targeted search yields up to a 10 -fold speed-up and 3 -fold reduction in structural error relative to GES" seems inappropriate in abstract, without a clear scope and simulation setup.

---

> ### Author Rebuttal · Authors · 2025-07-30
>
> Thank you for your thoughtful and comprehensive feedback. We believe some concerns may stem from a lack of clarity regarding technical aspects of our work. We hope you will reconsider the paper in light of the clarifications below.
>
> # W1
> 1.  *Prop 1 complete?* We appreciate the question and will revise the text to better motivate our heuristics.
>     1. The converse of Prop. 1 does hold. Prop. 1 shows which conditional independencies (CIs) can be inferred from the scoring operations that GES already conducts: a fact that LGES uses. Its converse shows which scoring operations can be conducted to test for certain CIs.
>     2. We do not check for the existence of an arbitrary separating set (as in PC). Local consistency only relates score changes to CIs with conditioning sets of the form $pa(Y,G)$ for some $G$ in $\mathcal{E}$.
> 2. *Heuristics reflect Prop 1?*
>       1. ConservativeInsert searches exactly for a score-decreasing Insert per Prop 1.
>       2. SafeInsert may not always detect such an $X-Y$ separation, since it chooses *one* Insert and checks if it decreases the score (details in Examples 2 and 3).
>     3. Both **ConservativeInsert and SafeInsert often successfully detect non-adjacencies in practice**, as suggested by their improvement on GES in Exp 5.1.
>
> # W2
> *LGES conducts more tests? Why is LGES faster?*
>  Thank you for raising this point; we will clarify it in the paper. **Both LGES variants perform no more (often fewer) scoring operations per step than GES**.
> 1. *"more robust decision… more tests… now all $\mathbf{T}$ need to be tested"* GES already tests all $X,Y$, and $\mathbf{T}$ to find the best Insert. In contrast, SafeInsert and ConservativeInsert stop early after finding a score-decreasing Insert. So, LGES is faster per step.
> 2. LGES is also faster overall due to its progression; it adds fewer edges, reducing future operator counts (exponential in the node degree).
>
> # W3
> *Deletion phase not optimized accordingly?*
> A score-decreasing Delete implies dependence under some conditioning set, not necessarily all. A true adjacency implies dependence given all conditioning sets. So, skipping all Deletes due to one score decrease would compromise soundness and risk admitting many false adjacencies. A true adjacency would imply all $X-Y$ Deletes are score-decreasing, so GES/LGES will never delete $X-Y$. Experiments support this: **false non-adjacencies are rare** (Fig D.1.1 (f)).
>
> # W4
> *Relevance of interventions/prior knowledge?*
> Algorithms that can use prior knowledge and interventional data have significant practical implications. In Experiment 5.2, prior knowledge can improve both accuracy and scalability. Interventional data improves identifiability [1] and is increasingly available in domains like biology due to novel experimental techniques, e.g., perturb-seq.
> 1. *“run plain GES…on all variables including the domain index.”*
>     1. If this method treats the domain index as a categorical root variable, then it may not be sound:
>         1. Data from different interventions is not identically distributed,  a core assumption in most causal discovery algorithms (including GES). This motivated GIES [1].
>         2. More edges can be oriented using interventional data, and interventional MECs tend to be smaller than observational MECs. However, this method would only learn the observational MEC over the variables plus domain index.
>     2. If you are referring to the use of F-nodes as intervention indicator variables [9, 10], this would be a valid but non-trivial extension of GES.
>     3. If we misunderstand your suggestion, we would appreciate a reference so we can better address the comparison.
> 2. *“Is this idea new…how do other score based variants incorporate background knowledge?”* Thank you for the valuable question. We will add this to the Related Works. To our knowledge, using priority constraints to guide the search is new to score-based methods. Other approaches include:
>     1. Knowledge-Guided GES [3], which initializes the search to user-provided edges, and strictly avoids insertions of forbidden edges. It is not theoretically sound if the user’s knowledge is imperfect. Even if the knowledge is perfect, it may not be sound due to the restriction on Insert operators.
>     2. Incorporating the prior into the score [4, 5], which scores the graph using a prior based on background knowledge (in addition to data fit). However, this has not been applied to theoretically sound causal discovery.
>     3. Temporal ordering [6] uses a user-provided ordering to prune the search space, though its guarantees are unclear.
> 3. *“technical discussions … too short in body.”*  With the additional content page if accepted, we are open to expanding and prioritizing these sections given reviewer consensus.
>
>
> # W5
> *Other GES variants?*
> We appreciate the suggestion and will organize a cleaner appendix on this point.
> 1. **SGES**. We could not find a public implementation for comparison.
> 2. **FGES**. FGES avoids inserting $X-Y$ edges if $X,Y$ are  marginally independent. This means FGES may not be theoretically sound (noted in Sec A.2). FGES also underperforms GES in time and accuracy [2], though its Java implementation prevents fair runtime comparisons.
> 3. **XGES**.
>     1. To facilitate a fair comparison and shared implementation across GES variants, we implemented the XGES-0 [5] heuristic of prioritizing deletions over reversals over insertions in our LGES Python code. The accuracy on average degree 2 ER graphs with up to 150 variables was: GES < XGES-0 < LGES (Safe) < LGES (Safe) + XGES-0 < LGES (Cons) < LGES (Cons) + XGES-0. The time: reverse of accuracy, but LGES (Cons) and LGES (Safe) + XGES-0 swapped.
>     2. To elaborate, the XGES-0 heuristic modestly improved the accuracy and runtime of each algorithm. This is consistent with [5], comparing GES-r (our GES) and XGES-0. **Both LGES variants (even w/o the XGES-0 heuristic) significantly outperform XGES-0**.
>     3. XGES extends XGES-0 by forcing edge deletions and restarts. This causes a ~10x slowdown relative to XGES-0 (Fig. 4 [5]), making it less practical for integration with LGES in high-dimensional settings.
> 4. **GES with repeats**. We ran GES/LGES with repeated applications of the phases  (proposed in [1] and by reviewer PUwW).  This modestly improved accuracy of GES/LGES, but **both LGES variants (even w/o repeats) significantly outperform GES w/ repeats.**
>
> **SHD**
> | Method/Variables| 25| 50  | 100 | 150 |
> |-|-|-|-|-|
> |GES| 4.38 ± 8.45    | 10.00 ± 9.54   | 31.78 ± 8.59   | 61.78 ± 10.26  |
> |XGES-0 | 3.04 ± 6.85    | 9.14 ± 8.38    | 30.66 ± 7.65   | 59.74 ± 10.10  |
> |LGES (Safe)| 3.62 ± 7.59    | 8.14 ± 9.48    | 25.80 ± 9.06   | 49.69 ± 10.02  |
> |XGES-0 + LGES (Safe)| 2.58 ± 5.80    | 7.14 ± 7.92    | 24.41 ± 7.97   | 47.42 ± 9.19   |
> |LGES (Cons)| 3.06 ± 6.91    | 4.94 ± 6.58    | 15.39 ± 8.82   | 29.94 ± 9.45   |
> |XGES-0 + LGES (Cons)|**2.56 ± 5.95**|**4.42 ± 5.70**|**14.66 ± 7.91**|**26.95 ± 6.88**|
>
> **Time**
> | Method/Variables| 25 | 50|100 |150|
> |-|-|-|-|-|
> | GES| 4.67 ± 1.89    | 58.22 ± 15.05  | 841.76 ± 102.80 | 4509.60 ± 1047.57 |
> | XGES-0| 3.85 ± 0.96    | 50.69 ± 10.94  | 759.37 ± 92.51  | 4043.42 ± 332.04  |
> | LGES (Safe)| 1.28 ± 0.84    | 8.81 ± 3.91    | 69.71 ± 22.33   | 251.39 ± 65.19    |
> | LGES (Cons) | 1.17 ± 0.59    | 8.39 ± 3.68    | 66.92 ± 21.82   | 240.40 ± 62.26    |
> | XGES-0 + LGES (Safe)| 1.07 ± 0.51    | 7.40 ± 3.16    | 55.44 ± 17.30   | 208.16 ± 51.64    |
> | XGES-0 + LGES (Cons)| **1.06 ± 0.57**    | **7.06 ± 2.99**    | **53.33 ± 17.18**   | **198.33 ± 49.03**    |
>
>
> # W6
> *SafeInsert details?*
> 1. “$nd_Y^G$ undefined?” Thank you for the catch. We'll define this as the non-descendants of $Y$ in $G$ in the prelim.
> 2. “SafeInsert...only one DAG from the CPDAG” Yes. SafeInsert picks one DAG $G$ in $\mathcal{E}$ and checks if  $X\perp Y\mid pa(Y,G)$. If so, we skip $X-Y$ Inserts. While we may skip a higher-scoring Insert, it's undesirable regardless, since $X,Y$ are separable. If $X \not\perp Y\mid pa(Y,G)$, we proceed as in GES. Using one $G$ for all $X,Y$ pairs ensures soundness.
> 3. “Unchanged score... impossible“ Correct; local consistency gives a strict score change in both directions.
> 4. “SafeInsert similar to GES” Not exactly. SafeInsert avoids adding $X-Y$ if it detects a separating set in its chosen $G$, even if such a separation isn't given by other DAGs in $\mathcal{E}$. GES may still add $X-Y$ if another DAG gives a higher score. We will clarify this in Ex. 2-3.
>
> [1] A. Hauser and P. Bühlmann. Characterization and greedy learning of interventional Markov equivalence classes of directed acyclic graphs. JMLR, 2012.
>
> [2] A. Nazaret and D. Blei. Extremely greedy equivalence search. UAI, 2024.
>
> [3] U. Hasan and M. O. Gani. Optimizing data-driven causal discovery using knowledge-guided search. arXiv preprint arXiv:2304.05493, 2024.
>
> [4] M. J. Bayarri, J. O. Berger, W. Jang, S. Ray, L. R. Pericchi, and I. Visser. Prior-based Bayesian information criterion. Statistical Theory and Related Fields, 2019.
>
> [5] N. Angelopoulos and J. Cussens. Bayesian learning of Bayesian networks with informative priors. Annals of Mathematics and Artificial Intelligence, 2008.
>
> [6] A. C. Constantinou, Z. Guo, and N. K. Kitson. The impact of prior knowledge on causal structure learning. Knowledge and Information Systems, 2023.
>
> [7] J. I. Alonso, L. de la Ossa, J. A. Gamez, and J. M. Puerta. On the use of local search heuristics to improve GES-based Bayesian network learning. Applied Soft Computing, 2018.
>
> [8] X. Liu, X. Gao, X. Ru, X. Tan, and Z. Wang. Improving greedy local search methods by switching the search space. Applied Intelligence, 2023.
>
> [9] K. Yang, A. Katcoff, and C. Uhler. Characterizing and learning equivalence classes of causal DAGs under interventions. ICML, 2018.
>
> [10] M. Kocaoglu, A. Jaber, K. Shanmugam, and E. Bareinboim. Characterization and learning of causal graphs with latent variables from soft interventions. NeurIPS, 2019.

---

> > ### Comment · Reviewer_Bizy · 2025-08-05
> >
> > Thank the authors for the detailed response. My concerns are addressed. I have increased my score accordingly.

---

> > > ### Author Response · Authors · 2025-08-05
> > >
> > > Thank you once again for your instructive comments, and for taking the time to reconsider our work in this productive exchange.

---

### Decision · Program_Chairs · 2025-09-17

**Decision:**

Accept (poster)

**Comment:**

This paper introduces Less Greedy Equivalence Search (LGES), which improves upon the classic Greedy Equivalence Search (GES) method by choosing edge orientations less greedily, instead of greedily maximizing the score.

In terms of strengths, the reviewers unanimously recognized the importance of the problem being studied and the significance of the new insights that this work offers. The writing quality of the paper (e.g., the illustrative and motivating examples) was also praised. Their I-Orient algorithm is also the first asymptotically correct score-based procedure for learning from interventional data that can scale to graphs with more than a hundred nodes.

There were a number of issues raised in the reviews, which the authors successfully responded to. We would request the authors to include in their revision the data and discussion from the new experiments they conducted for the rebuttal.